# Multi-decadal atmospheric and marine climate variability in southern Iberia during the mid- to late- Holocene

Julien Schirrmacher[1,2], Mara Weinelt[1], Thomas Blanz[3], Nils Andersen[2], Emília Salgueiro[4,5], Ralph R. Schneider[1,2,3]

[1]CRC 1266, Christian-Albrechts-Universität, Kiel, 24118, Germany
[2]Leibniz-Laboratory for Radiometric Dating and Stable Isotope Research, Christian-Albrechts-Universität, Kiel, 24118, Germany
[3]Department of Geosciences, Christian-Albrechts-Universität, Kiel, 24118, Germany
[4]Div. Geologia e Georecursos Marinhos, Instituto Português do Mar e da Atmosfera (IPMA), Lisbon, 1749-077, Portugal
[5]CCMAR, Centro de Ciências do Mar, Universidade do Algarve, Campus de Gambelas, 8005-139 Faro, Portugal

*Correspondence to*: Julien Schirrmacher (jschirrmacher@leibniz.uni-kiel.de)

**Abstract.** To assess the regional multi-decadal to multi-centennial climate variability along the southern Iberian Peninsula during the mid- to late- Holocene records of paleo-environmental indicators from marine sediments were established for two sites in the Alboran Sea (ODP-161-976A) and the Gulf of Cadiz (GeoB5901-2). High-resolution records of organic geochemical properties and planktic foraminiferal assemblages are used to decipher precipitation and vegetation changes as well as hydrological conditions with respect to Sea Surface Temperature (SST) and marine primary productivity (MPP). As a proxy for precipitation change records of plant derived n-alkane composition suggest a series of five distinct dry episodes in southern Iberia at $5.4 \pm 0.3$, from 5.1 to $4.9 \pm 0.1$, from 4.8 to $4.7 \pm 0.1$, from 4.4 to $4.3 \pm 0.1$ and, at $3.7 \pm 0.1$ cal. ka BP. During each dry episode the vegetation suffered from reduced water availability. Interestingly, the dry phase from 4.4 to $4.3 \pm 0.1$ cal. ka BP is followed by a rapid shift towards wetter conditions revealing a more complex pattern than was described for the 4.2 ka event in other regions. The series of dry episodes as well as closely connected hydrological variability in the Alboran Sea were probably driven by North Atlantic Oscillation (NAO)-like variability. In contrast, surface waters in the Gulf of Cadiz appear to have responded more directly to North Atlantic cooling associated with Bond Events. In particular, during Bond Events 3 and 4 a pronounced increase in seasonality with summer warming and winter cooling is found.

## 1 Introduction

Holocene climate has been considered to be fairly stable in comparison with the large and abrupt climatic changes during the last glacial and deglacial (Martrat et al., 2007; Rasmussen et al., 2014). For the Mediterranean realm, generally long-term trends in Sea Surface Temperature (SST) cooling (Kim et al., 2004; Martrat et al., 2014) and continental aridification (Fletcher

and Sánchez Goñi, 2008; Ramos-Román et al., 2018a) are described, superimposed several short cold and dry perturbations, for example the 8.2 ka event or the North Atlantic Bond Events (Alley et al., 1997; Bond et al., 1997; Mayewski et al., 2004). Another prominent Holocene climate perturbation is the 4.2 ka event. This event, considered to have had global impact, is associated with generally colder and wetter conditions over the North Atlantic as well as Northern and Central Europe and is

broadly coincident with the onset of Neoglacial glacier advances in Scandinavia and the Alps (Bakke et al., 2010; Le Roy et al., 2017). Additionally, intense droughts have been ascribed to this event for the mid-latitudes in Northern America and Eurasia including the Eastern and Central Mediterranean (Booth et al., 2005; Cheng et al., 2015; Jalut et al., 2000; Magny et al., 2013). In the Western Mediterranean the 4.2 ka event is so far insufficiently resolved in existing marine and terrestrial archives. Several marine archives reveal even contrasting (dry vs. wet) signals (Weinelt et al., 2015). Concerning terrestrial

records, a dry event at 4.2 cal. ka BP is suggested for southern Iberia (Schröder et al., 2018; Walczak et al., 2015) while relatively wet conditions were inferred for northern Morocco (Zielhofer et al., 2018).

Similar to the manifestation of the 4.2 ka event in the Western Mediterranean region, the driving mechanism(s) for it are not well understood. Many studies suggest North Atlantic Oscillation (NAO)-like atmospheric variability as potential driver of climatic change during the mid- to late- Holocene in the area (e.g. Deininger et al., 2017; Wassenburg et al., 2016). The NAO

is the dominant driver for the modern precipitation distribution in the Western Mediterranean and particularly active during winter (Hurrell, 1995). Another potential driving mechanism not only of the oceanic variability are the cyclical Bond Events (Bond et al., 1997), which are believed to weaken the thermohaline oceanic circulation and associated northward heat transport, thus, responsible for cooling of the northern hemisphere (Bond et al., 2001; Wanner et al., 2011).

Here, we explore the mid- to late- Holocene climate development in southern Iberia on the basis of two marine sediment cores

from the Gulf of Cadiz (GeoB5901-2) and the Alboran Sea (ODP-161-976A). Both sites are analysed for changes in terrestrial vegetation and precipitation as well as for changes in seasonal and annual SST and marine primary productivity (MPP). For that purpose, terrestrial n-alkane concentrations and the Norm33 n-alkane ratio from higher plant leaf-waxes are used to decipher the continental climate change. For the marine conditions, analyses of the alkenone-derived SST index ($U_{37}^{K'}$) and planktic foraminiferal assemblages applying the Modern Analogue Technique (MAT) were used to reconstruct annual mean

and seasonal changes in SST. Changes in MPP are based on fluctuations in the content of alkenones in the bulk sediment. Altogether, this study aims to provide new insights into the temporal and spatial manifestation of the 4.2 ka event as well as to discuss potential driving mechanisms by linking terrestrial and marine surface climate variability in southern Iberia and adjacent oceans.

## 1.1 Study area

The Iberian Peninsula is influenced by the major Atlantic and Mediterranean atmospheric regimes (Lionello, 2012). Today, the Atlantic climate is typically marked by relatively cool annual air temperatures and evenly distributed precipitation throughout the year. In contrast, the Mediterranean climate is characterized by pronounced seasonal contrasts with a rainy

winter season and a dry and hot summer season. In general, most of the precipitation at the Iberian Peninsula occurs during the winter season (Figure 1; Lionello, 2012). The spatial pattern of winter precipitation clearly reflects the two atmospheric regimes (Figure 1). The Atlantic regime, which spans along the western and northern coasts, is characterized by high precipitation of more than 800 mm during winter. It is associated with the westerly wind belt at the Iberian Peninsula (Zorita et al., 1992) and to a large extent controlled by the NAO winter conditions (Hernández et al., 2015; Hurrell, 1995). During

positive NAO conditions (i.e. a high pressure difference between the Azores High and the Icelandic Low) North Atlantic storm tracks and associated precipitation are directed towards northern Europe, while Iberia receives less rainfall. On the other hand, during negative NAO conditions (i.e. a low pressure difference) storm tracks are directed towards the Iberian Peninsula, which then experiences wetter winters. The pronounced difference in precipitation is also evident in the seasonal variability of the river discharges in southern Iberia. For example, the average discharge of the Guadalquivir is below about 100 hm$^3$ per month

from April to October and peaks during December and January with values above 400 hm$^3$ per month (Fernández-Delgado et al., 2007). Nonetheless, the central parts of the Iberian Peninsula as well as the eastern and southern coasts under influence of the Mediterranean climate remain relatively dry on average with precipitation less than 600 mm during winter (Figure 1).

A strong seasonal contrast is also evident in the surface ocean in the Gulf of Cadiz, which receives warm waters with the eastward Azores Current (AC) that turns into a poleward surface current (Iberian Poleward Current; IPC) along the western

Iberian coast during winter (Peliz et al., 2005). During summer, however, the surface current is directed southward and favours coastal upwelling along the western continental margin limiting the influence of the AC and, thus, causing relatively low SSTs (Figure 1; Haynes et al., 1993; Peliz et al., 2002). These opposite oceanographic currents limit the seasonality recognized in the SSTs in the Gulf of Cadiz. The seasonal difference at the core location of GeoB5901-2 is typically ~6 °C during modern times (Locarnini et al., 2013). Atlantic surface water passes the Strait of Gibraltar and enters the Alboran Sea. Within the

Alboran Sea it circulates in two anti-cyclonic gyres – the West Alboran Gyre (WAG) and the East Alboran Gyre (EAG) (Lionello, 2012). At the WAG relatively warm and fresh waters become upwelled almost continuously throughout the year (Sarhan et al., 2000). This upwelling is accompanied by an elevated MPP at the northern rim of the gyre (Minas et al., 1991). The modern seasonality in the Alboran Sea is very similar to the one in the Gulf of Cadiz (~6 °C).

## 2 Materials and methods

### 2.1 Sediment cores and sampling

For this study two marine sediment cores from the Alboran Sea and the Gulf of Cadiz were analysed. Sediment core ODP-161-976A (36°12.32' N; 4°18.76' W; 1108 m water depth) was retrieved in the Alboran Sea during JOIDES RESOLUTION cruise in 1995 (Comas et al., 1996). To achieve multi-decadal resolution, the section from 100.0 cm to 149.0 cm was continuously sampled at 0.5 cm distances in the IODP Core Repository at MARUM in Bremen (Germany). The analysed section consists of homogenous clayey and silty sediments, which appear slightly bioturbated. No hiatus or depositional event (e.g. turbidite) has been recognized throughout the section (Comas et al., 1996). Sediment core GeoB5901-2 (36°22.80' N; 7°04.28' W; 574 m water depth) was retrieved during METEOR cruise in 1999 in the Gulf of Cadiz (Schott et al., 2000). This sediment core, containing hemipelagic mud, was sampled on 0.5 cm resolution from 1.0 cm to 50.0 cm in the GeoB Core Repository at MARUM in Bremen (Germany). The samples used for organic geochemical analysis were freeze-dried and homogenized prior to analysis.

### 2.2 Age model

Age models of both cores are based on existing and new AMS[14]C dates, all measured at Leibniz Laboratory at Kiel University (Table 1). Published data of cores ODP-161-976A and GeoB5901-2 were taken from Combourieu Nebout et al. (2002) and Kim et al. (2004), respectively. In addition, we measured six new AMS[14]C dates on monospecific planktic foraminiferal samples of *Globigerinoides ruber* white + pink or *Globigerina bulloides* larger than 150 µm from sediment core ODP-161-976A as well as seven new dates from sediment core GeoB5901-2. A section from 116.25 cm to 124.75 cm in ODP-161-976A included three samples yielding the same AMS[14]C age (see Table 1). Since there is no evidence for strong bioturbation, hiatuses or, turbidites we assume a rather continuous sedimentation rate throughout the core and, thus, decided to take the date at 120.25 cm as the only age control point for this depth interval since it has been dated twice on two different planktic foraminifera. The dates at 116.25 cm and 124.75 cm were not considered for the final age model in order to avoid artificially induced calculations of extraordinary high sedimentations rates in this thin sediment increment. The final age models have been calculated using Bacon (Blaauw and Christen, 2011). During age model processing all AMS[14]C dates have been calibrated using the marine13 calibration curve including a global mean reservoir correction of 400 years (Reimer et al., 2013). Based on the AMS[14]C dates, we assume a relatively high accumulation rate for ODP-161-976A. Accordingly, we set the mean accumulation parameter of the model to 50 yr/cm and run the model for 2.5 cm thick sections to allow for a certain variability of the accumulation rates throughout the section. The accumulation rate of GeoB5901-2 is expected to be lower in the analysed section based on AMS[14]C data. Thus, the model parameters have been set to 100 yr/cm and 5 cm thick sections. In the case of sediment core ODP-161-976A the initial age model of Jiménez-Amat and Zahn (2015) was shifted significantly within the

studied section by about 700 years. The reason for this shift are the new AMS[14]C dates accomplished during this study emphasizing the need of a dense dating strategy. According to the final age model, sedimentation rates range between 9.8 and 50.0 cm/kyr in core ODP-161-976A and between 4.8 and 33.3 cm/kyr in core GeoB5901-2, respectively. With 0.5 cm sampling intervals decadal to multi-decadal resolution is achieved. The studied section of sediment core ODP-161-976A dates between

approximately 5.4 to 3.0 cal. ka BP exhibiting a temporal resolution between 10 to 57 years per sample with continuous high resolution between 5.4 and 4.6 cal. ka BP. The analysed section of sediment core GeoB5901-2 dates between approximately 6.2 and 1.8 cal. ka BP resulting in a temporal resolution between 15 and 104 years per sample. Age models and sedimentation rates for both sediment cores are shown in Figure 2 and 3, respectively.

**2.3 Organic geochemical analysis and calculations**

Lipids were extracted from the freeze-dried and finely ground sediment samples with an Accelerated Solvent Extractor (ASE-200, Dionex) at 100 bar and 100 °C using a 9:1 (v/v) mixture of dichloromethane (DCM) and methanol. After extraction samples were de-sulphured by stirring for 30 minutes with activated copper. The de-sulphured lipids were subsequently separated by silica gel column chromatography using activated silica gel (450 °C for 4 h) into neutral (hexane) and polar (DCM) fractions containing n-alkanes and alkenones, respectively. The neutral fraction was further separated using silver-

nitrate (AgNO$_3$) coated silica gel. Samples were then left at room temperature for approximately 24 hours for homogenisation. Afterwards, n-alkanes were analysed by gas chromatography (GC) using an Agilent 6890N gas chromatograph equipped with a Restek XTI-5 capillary column (30 m x 320 µm x 0.25 µm) and a Flame Ionization Detector (FID) at the Institute of Geosciences, Kiel University. Example chromatograms are provided in the supplement of this paper. n-Alkanes were identified by comparison of their retention times with an external standard containing a series of n-alkane homologues of known

concentration. On this basis, n-alkanes were also quantified using the FID peak areas calibrated against the external standard. For environmental interpretation the sum of terrestrial sourced, odd n-alkane homologues C$_{27}$ to C$_{33}$ is used. The mean analytical error (2σ) is 7.0 ng/ g sediment based on replicate analyses (n = 62).

Various studies used ratios among individual n-alkane homologues as environmental sensitive parameter, for example the Norm33 ratio (e.g. (Herrmann et al., 2016). The Norm33 ratio was calculated by the following equation:

Norm33 = C$_{33}$ / (C$_{29}$ + C$_{33}$)                                                                                    (1)

where C$_X$ is the peak area of the n-alkane with x carbon atoms in the chromatogram. The mean analytical error (2σ) is 0.004 based on replicate analyses (n = 62).

Alkenones were analysed on a multi-dimensional, double gas column chromatography (MD-GC) set up with two Agilent 6890 gas chromatographs. The compounds (C$_{37:2}$ and C$_{37:3}$) were quantified by calibration to an external standard. The alkenone

concentration is derived from the sum of the C$_{37:2}$ and C$_{37:3}$ isomers with a mean analytical error (2σ) of 6.9 ng/g sediment. The

alkenone unsaturation index ($U_{37}^{K'}$) was obtained by using the peak areas of the aforementioned compounds applying the equation of Prahl and Wakeham (1987):

$$U_{37}^{K'} = C_{37:2} / (C_{37:2} + C_{37:3})$$ (2)

where $C_X$ represents the respective peak area in the chromatogram. The $U_{37}^{K'}$ index was subsequently transferred into annual mean SST using the calibration of Müller et al. (1998):

$$SST\ (°C) = (U_{37}^{K'} - 0.044) / 0.033$$ (3)

The laboratory internal analytical error is approximately 0.12 °C, while the error of the calibration is 1.5 °C.

## 2.4 Planktic foraminiferal analysis and Modern Analogue Technique

For foraminiferal analysis sediment samples of approximately 10 cc were washed over 63 μm sieves, dried at 40 °C and, subsequently dry sieved for larger fractions. Planktic foraminifera assemblages were analysed in the size fractions >150 μm enabling the application of commonly used transfer techniques for SST reconstructions based on relative abundances of 26 taxonomic categories within the assemblage, following concept by Pflaumann et al. (1996). For reliable assemblage counts samples were dry split into aliquots of at least 300 specimens with a Kiel dry sample splitter. For SST estimates we used SIMMAX non-distance-weighted modern analogue technique, using a similarity index of >0.963 and based on 10 closest analogues (Pflaumann et al., 2003). Because the study sites are influenced by Atlantic and Mediterranean ocean circulation, we combined an updated North Atlantic core top database (Kucera et al., 2005a,b; Salgueiro et al., 2014) and the Mediterranean database (Hayes et al., 2005) with modern temperature at 10 m water depth taken from the World Ocean Atlas 1998 (Salgueiro et al., 2014). Seasonal temperatures are averaged for northern hemisphere summer (July to September) and winter (December to February). This method applied on overall 1212 core top samples from the Atlantic and the Mediterranean yields an root-mean square error predicted accuracy of ± 1.3 °C for summer and winter seasons.

## 3 Results

### 3.1 ODP-161-976A

*Terrestrial proxies.* The terrestrial n-alkane concentration varies between 67 and 714 ng/g sediment exhibiting no long-term trend across the covered time period from 5.4 to 3.0 cal. ka BP (Figure 3). Three distinct concentration minima below 200 ng/g sediment can be observed at 5.4 cal. ka BP, from 5.0 to 4.9 cal. ka BP and, from about 4.8 to 4.7 cal. ka BP. Less distinct minima are observed between 4.4 and 4.3 cal. ka BP and at about 3.7 cal. ka BP. A sharp increase towards high concentrations of up to 617 ng/g sediment is evident at about 4.2 cal. ka BP. n-Alkane concentrations appear to remain generally above 450 ng/g sediment until 3.8 cal. ka BP, when concentrations sharply decrease towards 209 ng/g sediment at 3.7 cal. ka BP.

Afterwards, n-alkane concentrations reveal a slightly increasing trend towards 3.0 cal. ka BP. The Norm33 ratio exhibits no trends and varies between 0.29 and 0.49 (Figure 3). Four sharp increases towards values greater than 0.42 can be recognized at about 5.4 cal. ka BP, from 5.0 to 4.9 cal. ka BP, from 4.8 to 4.7 cal. ka BP and, at about 4.6 cal. ka BP. Apparently, the Norm33 maxima parallel the n-alkane concentration minima between 5.4 and 4.7 cal. ka BP.

*Marine proxies.* The alkenone derived annual mean SST remains quite stable between 18.9 °C and 20.0 °C (Figure 3). No trends are visible in the analysed time period. Only weak maxima in the annual mean SST of 0.5 to 1.0 °C in amplitude are apparent at about 5.4, 3.3 and, 3.0 cal. ka BP as well as between 5.1 and 4.6 cal. ka BP at about 100 year intervals (Figure 4). All foraminifera-derived seasonal SST reconstructions suggest similar stable temperature conditions without any obvious trends between 5.4 and 3.0 cal. ka BP. Summer SSTs vary around $22.8 \pm 0.1$ °C with several cooling episodes of up to 1 °C at

about 5.0, 4.6, 4.3, 3.6, 3.3 and, 3.2 cal. ka BP. Winter SST generally vary around $15.0 \pm 0.1$ °C with weak warming episodes of up to 1 °C at about 5.0, 4.6, 4.3, 3.6, 3.3 and, 3.2 cal. ka BP. Thus, the winter SST maxima and summer SST minima are contemporaneous and result in a decreasing seasonal SST difference during these periods. Notably, the seasonal SST maxima or minima, respectively, do not parallel the SST maxima observed in the annual mean reconstruction. The alkenone concentration varies between about 224 and 440 ng/ g sediment and shows an increasing trend from 5.4 to 4.0 cal. ka BP

followed by a decreasing trend towards 3.0 cal. ka BP (Figure 4). These trends are superimposed by several minima from 5.4 to 5.3 cal. ka BP, around 5.2, 5.0, 4.8 cal. ka BP, from 4.7 to 4.6 cal. ka BP, at 3.8 cal. ka BP and, from 3.1 to 3.0 cal. ka BP. The alkenone concentration is to some degree correlated ($r = 0.61$) to the annual mean SST across the studied time period (see supplement) with warmer annual mean SSTs paralleled by decreased alkenones content at about 5.4, 5.2, 4.9, 4.7, 4.6 cal. ka BP and, from 3.1 to 3.0 cal. ka BP (Figure 4).

**3.2 GeoB5901-2**

     *Terrestrial proxies.* The terrestrial n-alkane concentration varies between 9 and 535 ng/g sediment and has an increasing trend towards younger ages between 5.4 and 3.0 cal. ka BP (Figure 3). During the studied time period at least three periods of decreasing n-alkane concentration below 100 ng/g sediment are found from 5.1 to 4.7 cal. ka BP, from 4.4 to 4.3 cal. ka BP and, at about 3.7 cal. ka BP. The oldest concentration minimum reveals a "W"-shaped pattern with increasing concentrations

of up to 110 ng/g sediment at about 4.9 cal. ka BP. An additional concentration minimum (46 ng/g sediment) at 5.2 cal. ka BP is only corroborated by a single data point and, thus, not robust. The Norm33 varies without any trends between 0.31 and 0.51 showing high amplitude maxima with values above 0.42 at about 5.0, 4.8, 4.3 and, 3.7 cal. ka BP (Figure 3). Notably, the Norm33 reveals maxima during periods of low terrestrial n-alkane concentration.

     *Marine proxies.* Alkenone derived annual mean SST vary without any trends between 5.5 and 3.0 cal. ka BP and reveal SSTs

around $20.4 \pm 0.3$ °C with the exception of a period from about 4.3 to 3.9 cal. ka BP, where SST vary around $21.6 \pm 0.6$ °C with maximum SST being 22.7 °C (Figure 5). Foraminifera-derived summer SST reveal a slight decreasing trend towards the

present between 6.2 and 1.8 cal. ka BP. Notably, between about 6.2 and 4.0 cal. ka BP the variability in summer SST is quite high with a mean of 22.4 ± 0.7 °C superimposed by several warm events of up to 23.9 °C that occur at 6.0, 5.8, 5.5, 4.7 and, 4.1 cal. ka BP. Also, the variability and the amplitudes of the warm events decrease towards the present. After 4.0 cal. ka BP summer SST vary around 21.7 ± 0.2 °C. Winter SST reveal a very similar pattern compared to the summer SST with high variability (16.0 ± 0.7 °C) until 4.0 cal. ka BP and less (16.8 ± 0.2 °C) afterwards. Overall, there is a warming trend from about 16 to 17 °C towards younger ages with major cooling episodes at 6.0, 5.8, 5.5, 4.7 and, 4.1 cal. ka BP. Winter SST decrease below 15.0 °C during most of these periods. The observed SST events in the seasonal reconstructions (summer and winter) are contemporaneous and result in an increasing seasonal difference of up to 9.3 °C during these events (Figure 5). Overall, the seasonal difference is decreasing towards younger ages from 6.4 to 4.7 °C. SST events within the seasonal reconstructions do not parallel any events in the alkenones derived annual mean SST. The warm period observed in the annual mean data between 4.3 and 3.9 cal. ka BP is not visible in the seasonal data. Also, the annual mean SST is very close to summer conditions, even exceeding those around 4.2 cal. ka BP.

### 3.3 Comparison of both sediment cores

In general, a good agreement between the n-alkane concentration in both sediment cores is apparent (Figure 3). This is corroborated by the high correlation coefficient (r = 0.82) for the period between 5.5 and 3.6 cal. ka BP. Since the correlation calculation is based on 100-year time slices (see supplement), there is not such a robust correlation in the younger part between 3.6 and 3.0 cal. ka BP (r = 0.04). Nonetheless, visual inspection implies a good agreement between both records when only the more pronounced minima and maxima are considered for this younger interval. The n-alkane concentration minima observed at about 4.3 and 3.7 cal. ka BP appear to be contemporaneous in both regions. Also, the two events from about 5.0 to 4.9 cal. ka BP and from 4.8 to 4.7 cal. ka BP in ODP-161-976A as well as the "W"-shaped event between 5.1 and 4.7 cal. ka BP in GeoB5901-2 correspond to a large extent. Only the drop towards low n-alkane concentrations between 5.1 and 5.0 cal. ka BP is offset by about 100 years (Figure 3). Bearing in the mind the chronological uncertainties around 5.0 cal. ka BP of ±110 years and ±160 years in the age model of GeoB5901-2 and ODP-161-976A, respectively, this offset likely is chronological bias. The Norm33 maxima of both sediment cores correspond well between 5.4 and 4.7 cal. ka BP (Figure 3). With respect to absolute SST values and trends in the temperature reconstructions, differences between both cores are larger than for the terrestrial biomarker records. The alkenone-based as well as the foraminifera-derived winter SST were about 1 °C warmer in the Gulf of Cadiz, while summer SST were about 1°C warmer in the Alboran Sea. The most apparent difference between both regions is noticeable in the seasonal warm/cold events, which do not agree in their timing, as is the same for the much more moderate events observed in the alkenones-based SST records. Also, the overall trend in the difference in seasonal SST estimates is not similar in both regions. While in the Gulf of Cadiz (GeoB5901-2) the seasonal difference increases (i.e.

summer warming and winter cooling), the seasonal difference in the Alboran Sea record (ODP-161-976A) record does not change significantly.

## 4 Discussion

### 4.1 Terrestrial climate conditions in southern Iberia

Terrestrial plants synthesize long chain n-alkanes in their leaves as coating for protection against water loss (Eglinton and Hamilton, 1967). These long chain n-alkanes are either eroded directly from the leaves by wind or deposited in the soils in autumn. Afterwards, n-alkanes are removed from the atmosphere or eroded from the soils by rain and are further transported into the marine realm via aeolian and riverine transport (e.g. Bird et al., 1995; Conte and Weber, 2002; Schreuder et al., 2018). The fraction of the riverine input in coastal areas is usually much higher compared to the aeolian input. Therefore, terrestrial
n-alkane concentrations have already been successfully applied to study the riverine input in marine settings in the Mediterranean (Abrantes et al., 2017; Cortina et al., 2016; Jalali et al., 2016; 2017). More specifically, Rodrigo-Gámiz et al. (2015) have shown from radiogenic isotopes that in the Alboran Sea the riverine dominates the aeolian input during the studied time period. Because of the dominant riverine transport mechanism, it is assumed that both proxies integrate spatially over the river catchment areas shown in Figure 1 and, thus, indicate climate conditions for the entire southernmost Iberian Peninsula.
The dominant catchment for sediment core GeoB5901-2 is the one of the Guadalquivir river draining into the Gulf of Cadiz. The smaller mountainous rivers draining the southern Sierra Nevada area are considered as more relevant for sediment core ODP-161-976A. A delivery of material from the Guadalquivir with the inflow of the Atlantic water through the Strait of Gibraltar cannot be excluded, though. Since the river discharge and also the plant growing season is strongly coupled to precipitation and the rainy season at the Iberian Peninsula is in winter (Figure 1; Lionello, 2012), the n-alkane proxies are
probably biased towards the winter season.

Consequently, intervals of very low n-alkane concentration in the studied cores are interpreted as periods of dry winters, which occurred in southern Iberia at $5.4 \pm 0.3$, from 5.1 to $4.9 \pm 0.1$, from 4.8 to $4.7 \pm 0.1$, from 4.4 to $4.3 \pm 0.1$ and, at $3.7 \pm 0.1$ cal. ka BP (Figure 3). Drier conditions at about 5.5 cal. ka BP well in phase with the end of the African Humid Period (deMenocal et al., 2000). This transition appears to be marked by an aridity event as evidenced by speleothem data from El Refugio Cave
and Grotte de Piste (Walczak et al., 2015; Wassenburg et al., 2016) and high charcoal concentration in Cabo de Gata between about 5.4 and 5.3 cal. ka BP. Also, a contemporaneous remarkable decrease in Mediterranean forest was noticed by Ramos-Román et al. (2018b) from 5.5 to 5.4 cal. ka BP. Schröder et al. (2018) describe a drought event centred at about 5.3 cal. ka BP in Lake Medina (SW Iberia). The dry phases between about 5.1 and 4.9 and from 4.8 to 4.7 cal. ka BP observed in this study are corroborated by regional speleothem records (Moreno et al., 2017; Wassenburg et al., 2016). A dramatic forest
decline occurred between 5.0 and 4.5 cal. ka BP in SE Iberia (Pantaléon-Cano et al., 2003) along with high charcoal

concentrations at Cabo de Gata indicating more frequent wild fires (Burjachs and Expósito, 2015). Moderate forest declines are found in pollen records from the Alboran Sea and Elx sequence (Figure 6; Burjachs and Expósito, 2015; Fletcher and Sánchez Goñi, 2008). The following period between 4.4 and 3.8 cal. ka BP comprising the 4.2 ka event is discussed in more detail in chapter 4.2. The youngest dry phase found in this study at about 3.7 cal. ka BP has also been inferred from pollen data

at Elx and Villaverde (Burjachs and Expósito, 2015; Carrión et al., 2016), while pollen data from the Alboran Sea indicate a moderate forest decline (Fletcher and Sánchez Goñi, 2008). Additional evidence for a severe dry phase around 3.7 cal. ka BP again stems from speleothem data (Moreno et al., 2017; Walczak et al., 2015).

Notably, in the GeoB5901-2 data all dry episodes are paralleled by Norm33 maxima, while in the ODP-161-976A record the dry phases at about 5.4, 4.9 and, 4.7 cal. ka BP coincide with Norm33 maxima (Figure 3). A tendency towards higher n-alkane

chain lengths, as revealed by increasing Norm33 values, is an indication for warmer and/or drier environmental conditions (Bush and McInerney, 2013; Leider et al., 2013; Rommerskirchen et al., 2006). A local study from southern Iberia further shows that increasing chain lengths are a function of decreasing water availability (García-Alix et al., 2017). Consequently, the Norm33 maxima observed in this study indicate high water stress for the plants in relation to the dry climate episodes. This is supported by Sierra Nevada bog sediments Borreguil de la Virgen and Borreguil de la Caldera (Figure 6; García-Alix et al.,

2018), which also show Norm33 maxima during such dry episodes. It has further been shown that Mediterranean forests declined while scrubs expanded in southern Iberia during the respective dry phases (Fletcher et al., 2007; Ramos-Román et al., 2018a;b). Accordingly, the dry episodes caused a noticeable response in the vegetation, which suffered from decreasing water availability. This further implies that the vegetation in southern Iberia at that time was very sensitive to winter precipitation changes.

**4.2 The 4.2 ka BP event**

The manifestation of the 4.2 ka event across the Mediterranean is currently intensively debated with a particular focus on the western part. In the Eastern and Central Mediterranean many studies suggest a dry phase in the time window between 4.4 and 3.8 cal. ka BP (e.g. Calò et al., 2012; Cheng et al., 2015; Finné et al., 2017; Zanchetta et al., 2016). In the Western Mediterranean evidence for a dry phase associated with the 4.2 ka event comes from northern Algeria (Ruan et al., 2016) and

pollen data from southern Iberia (Ramos-Román et al., 2018a). Furthermore, a drastic forest opening in SE Iberia is indicated from pollen sequences at Cabo de Gata around 4.4 cal. ka BP and at Elx at about 4.3 cal. ka BP (Figure 6; Burjachs and Expósito, 2015). Additional indications for a dry phase coinciding with the 4.2 ka event come from lithological analyses in SE Iberia (Navarro-Hervás et al., 2014), a hiatus in pollen data from SW Iberia (Schröder et al., 2018) and, speleothem data (Moreno et al., 2017; Walczak et al., 2015; Wassenburg et al., 2016).

Compared to the prolonged dry phase between about 4.4 and 3.8 cal. ka BP recorded in speleothems from the Italian Peninsula and Algeria (Ruan et al., 2016; Zanchetta et al., 2016), our data indicate a more complex environmental pattern associated with

the 4.2 ka event. The terrestrial n-alkane concentrations suggest a dry period from about 4.4 to 4.3 cal. ka BP followed by a rapid return to wet conditions at about 4.2 cal. ka BP, which lasted until approximately 3.8 cal. ka BP. The Norm33 data, further, shows a peak in the GeoB5901-2 at about 4.3 cal. ka BP indicating that plants in the Guadalquivir basin were suffering from water stress. Contrastingly, the intermediate drop in n-alkane concentration in ODP-161-976A from the Alboran Sea is not paralleled by a Norm33 maximum. This might imply that the dry episode between about 4.4 and 4.3 cal. ka BP was more severely manifested in the lowlands of the Guadalquivir basin compared to the high-altitudes of the Sierra Nevada. Altogether, our reconstructions show that the onset of the dry phase at about 4.4 cal. ka BP is in line with proxy data from the Central Mediterranean, while at about 4.2 cal. ka BP a rapid return to wet conditions lasting until about 3.8 cal. ka BP contrasts the picture of a prolonged dry phase associated with the 4.2 ka event. This interpretation is corroborated by dry conditions at 4.4 cal. ka BP and wet conditions at 4.2 cal. ka BP found in ostracod shells from northern Morocco (Zielhofer et al., 2017; 2018).

## 4.3 Hydrological conditions in the Gulf of Cadiz and the Alboran Sea

In contrast to the terrestrial climate variability our marine reconstructions suggest fairly stable conditions between 5.4 and 3.0 cal. ka BP in the Alboran Sea and between 6.2 and 1.8 cal. ka BP the Gulf of Cadiz. The maximum range in the annual mean values based on alkenones is about 1 °C and, thus, within the calibration uncertainty. The low temporal resolution of previously studied cores from the area (below about 115 and 250 years for annual mean and seasonal data, respectively) further hampers the comparison with here observed SST events. Therefore, a careful interpretation of the alkenones-based SST results is required.

Alkenone derived SSTs from sediment core ODP-161-976A suggest annual mean temperatures between 18.9 and 20.0 °C (Figure 4), which exceed the modern values of about 18.0 °C (Locarnini et al., 2013). An overall cooling trend over the course of the mid- to late- Holocene in response to decreasing insolation (Figure 5) is known on regional as well as on global scale (e.g. Cacho et al., 2001; NGRIP-Members, 2004), thus corroborating warmer SSTs during the mid- Holocene compared to modern SSTs. Annual mean SST values around 19 °C at that time were also reconstructed by other studies from the Alboran Sea (Ausín et al., 2015; Cacho et al., 2001; Martrat et al., 2014; Pérez-Folgado et al., 2003; Rodrigo-Gámiz et al., 2014). Moreover, Cacho et al. (2001) described a cold SST event of about 1.5 °C in the Alboran Sea between 5.94 and 4.75 cal. ka BP. Our data from the Alboran Sea shows no cooling event of similar magnitude between 5.4 and 4.7 cal. ka BP. Bearing in mind the error of the calibration there might be a cooling at about 5.3 cal. ka BP in the order of just 0.5 °C, but surface water conditions between 5.4 and 4.7 cal. ka BP appear to have even been warmer compared to the period afterwards. In the Gulf of Cadiz alkenones-based SST values vary between 20.0 to 22.7 °C and are warmer compared to recent annual mean temperatures (18.7 °C; Locarnini et al., 2013). Higher SSTs during the mid-Holocene might be partly a consequence of higher insolation during this period. However, previous alkenone analyses of Kim et al. (2004) on the same sediment core yielded SST values

between 19 and 20 °C using a different SST calibration of Prahl et al. (1988), which would result in slightly cooler SSTs for our data as well. Additionally, annual mean SSTs around 20 °C in Gulf of Cadiz were also found by Cacho et al. (2001).

*Summer temperatures.* Reconstructed summer SST based on foraminiferal assemblage variations vary around 22.8 °C in the Alboran Sea and around 22.4 and 21.7 °C before and after approximately 4.0 cal. ka BP, respectively, in the Gulf of Cadiz.

Alboran Sea summer SSTs are exceeding modern ones (21.4 °C; Locarnini et al., 2013) likely due to higher insolation forcing during the mid- Holocene. This is even better demonstrated by the summer SST record in the Gulf of Cadiz. Here, summer SST estimates are progressively cooling towards mean summer SSTs around 21.7 °C, which perfectly agree with modern summer temperatures in the Gulf of Cadiz (21.7 °C; Locarnini et al., 2013). Reconstructed Alboran Sea summer SSTs are supported by Rodrigo-Gámiz et al. (2014), who estimated summer SSTs varying around 22 °C between 5.5 and 3.0 cal. ka BP.

Pérez-Folgado et al. (2003) found even warmer summer SSTs between 24 and 25 °C in the Alboran Sea. This difference may be a result of the Pérez-Folgado et al. (2003) data calibrated against August temperatures only, while our method used the mean of the summer season (July – September). These authors also found a 0.5 °C cooling between about 5.0 and 4.5 cal. ka BP, when our data implies two cold events of approximately 1 °C  at about 5.0 and 4.6 cal. ka BP. In the Gulf of Cadiz, in sediment core MD99-2339 (Salgueiro et al., 2014), summer SSTs vary between approximately 22 and 24 °C, exhibiting a

cooling trend towards the present, which is in harmony with our summer SST reconstructions deduced from GeoB5901-2.

*Winter temperatures.* Winter SSTs in the Alboran Sea vary around ca. 15.0 °C between 5.4 and 3.0 cal. ka BP and agree with modern conditions (15.4 °C; Locarnini et al., 2013). Moreover, winter SSTs (February) around 14.5 °C during the mid- to late-Holocene reported by Pérez-Folgado et al. (2003) support our results. Winter SSTs between 16 and 17 °C between 6.2 and 1.8 cal. ka BP in the Gulf of Cadiz are also in good agreement with the modern winter SSTs of 16.0 °C (Locarnini et al., 2013).

Slightly colder winter SSTs between 15 and 16 °C were reconstructed from dinocyst assemblages in the neighbouring core MD99-2339 (Penaud et al., 2016).

## 4.4 Potential drivers of terrestrial and oceanic climate variability

Under modern conditions the NAO is responsible for much of the variability in winter precipitation at the Iberian Peninsula (Hurrell, 1995; Zorita et al., 1992). Many paleo-climatic studies provide evidence for NAO-like climate variability since the

mid- Holocene (Abrantes et al., 2017; Deininger et al., 2017; Olsen et al., 2012). We, thus, compare the observed sequence of dry climate episodes to a reconstruction of NAO-like variability from Lake SS1220 in Greenland (Figure 4; Olsen et al., 2012). All dry phases observed in this study can be associated with positive excursions in the NAO reconstruction. Moreover, it is evident that a wetter period between about 4.2 and 3.8 cal. ka BP coincides with a period of more stable, less positive NAO-like conditions. Accordingly, we conclude that the dry phases observed between 5.4 and 3.0 cal. ka BP in southern Iberia were

caused by more dominant positive NAO-like conditions. Yet, other atmospheric circulation patterns like the Scandinavian and the East Atlantic patterns also influence precipitation variability at the Iberian Peninsula (Abrantes et al., 2017; Hernández et

al., 2015). This may explain, why our records do not reveal dry episodes between 4.7 and 4.5 cal. ka BP and at about 3.1 cal. ka BP despite the NAO reconstruction suggests a strong positive circulation mode for these times. Interestingly, at around 5.2, 4.6 and, 3.0 cal. ka BP we found contemporaneous minima in the MPP and maxima in annual mean SSTs in the Alboran Sea (Figure 6). This linkage is also visible in the dry phases at about 5.4 cal. ka BP and between 5.0 and 4.8 cal. ka BP, while during those at about 4.3 and 3.7 cal. ka BP the warming of the annual mean SSTs was more pronounced. This linkage has previously been described by Ausín et al. (2015) since 7.7 cal. ka BP in the Alboran Sea and can also be related to NAO forcing. Currently, the inflow of the Atlantic Waters through the Strait of Gibraltar is strong under negative NAO conditions and, thereby, its flow within the Alboran Sea is shifted southwards. This promotes intensified upwelling of cold and nutrient rich waters, which also result in higher MPP (Sarhan et al., 2000). During positive NAO conditions, on the other hand, the inflow of the Atlantic Waters is weakened resulting in a more stable water column with limited upwelling and, thus, low MPP (Sarhan et al., 2000). Despite the calibration uncertainty of the annual mean SST reconstructions a similar mechanism is also indicated by our data during the mid- to late- Holocene. Accordingly, we conclude that a NAO-like variability was coupled to the terrestrial variability in southern Iberia as well as for the surface water variability in the Alboran Sea between 5.4 and 3.0 cal. ka BP.

It has further been suggested that NAO-like variability in the past was responsible for oceanic changes in the subtropical Atlantic such as the position of the Azores Front (Goslin et al., 2018; Repschläger et al., 2017) as well as for increased upwelling and related SST changes along the western Iberian margin (Abrantes et al., 2017). In more detail, Abrantes et al. (2017) suggest that warm winters and cool summers during the Medieval Climate Anomaly were related to positive NAO-like conditions. For the time interval between 6.2 and 1.8 cal. ka BP we cannot confirm a general link between the NAO and the hydrological conditions in the Gulf of Cadiz. Interestingly, we observe two notable differences between the hydrological conditions in the Gulf of Cadiz and the Alboran Sea: (1) the summer/winter SST variability in the Gulf of Cadiz is opposite resulting in a greater seasonality during seasonal SST events, while seasonality appears to decrease during summer/winter events in the Alboran Sea within methodological error and (2) the observed seasonal warming/ cooling events in both adjacent basins are not contemporaneous. This suggests different mechanisms driving the hydrological variability in the Gulf of Cadiz and the Alboran Sea. The pronounced seasonal SST variability in the Gulf of Cadiz is in general agreement with the stacked Ice-Rafted Debris (IRD) records for the North Atlantic (Bond et al., 2001; Figure 5). This implies that larger seasonal SST contrasts in the Gulf of Cadiz relate to periods of enhanced iceberg discharge in the North Atlantic during Bond Events 3 and 4. During Bond Events cold polar water masses have been advected as far south as the latitude of Britain (Bond et al., 1997). Our data suggests that this regime resulted in colder winter temperatures in the Gulf of Cadiz as well. Moreover, the Bond Events likely weakened the deep water formation in the North Atlantic by limiting the oceanic heat transport towards the north (Bond et al., 2001; Wanner et al., 2011). Repschläger et al. (2017) further proposed that early- Holocene freshwater forcing from the Laurentide ice sheet may have resulted in a weak subsurface heat transport from the Subtropical Gyre (STG) towards

the North Atlantic and, subsequently, in a subsurface heat storage within the STG. Following these mechanisms, we hypothesize that during Bond Events 3 and 4 the northward heat transport was blocked due to freshwater forcing in the North Atlantic, which in turn results in a heat storage within the STG. During summer, when the Inter-Tropical Convergence Zone (ITCZ) moved northwards, these heated water masses probably reached the Gulf of Cadiz via the Azores Current resulting in

warmer summer temperatures. These warmer water masses could also reach the Alboran Sea through the Strait of Gibraltar, but are not recorded within our data because summer temperatures in the Alboran Sea were generally warmer by about 1 °C during the studied period. Interestingly, it seems that annual mean SSTs in the Gulf of Cadiz warmed by up to 2 °C during Bond Event 3 (Figure 5). This might imply rather a growing season shift of the alkenones producing coccolithophores from spring to summer rather than a warming of the annual mean SSTs. Such a reaction of the coccolithophores might be also

indicated to a lesser extent during Bond Event 2 when annual mean SSTs warmed by approximately 1 °C at about 3.2 cal. ka BP. Our data does not cover the whole interval, though. Also, Bond Event 4 is not entirely covered so that this interpretation remains hypothetical. But all in all, the influence of cold water from the North Atlantic during winter along with the influence of warmer subtropical water masses during summer likely resulted in a pronounced seasonality during Bond Events 3 and 4 (Figure 5). Notably, the amplitude of the seasonal events and their according high seasonality as well as the overall seasonality

in the Gulf of Cadiz were decreasing towards the present. It is, moreover, interesting to note that Bond Event 2 is not visible in our seasonal SST records at all (Figure 5). The general decrease in seasonality in the Gulf of Cadiz can be attributed to decreasing summer insolation. We hypothesize that this is also true for the decreasing amplitude of the seasonal SST events. During Bond Event 4, when summer insolation was high, the more northward position of the ITCZ during summer allowed for a stronger inflow of warmer subtropical water masses into the Gulf of Cadiz. On the other hand, during winter the ITCZ

was much further south compared to its present position, allowing enhanced southward flow of the colder water masses from the north. During Bond Event 3 the summer and winter positions of the ITCZ were already less extreme, thus, weakening the influence of either warm and cold water masses during summer or winter, respectively. Later, during Bond Event 2 seasonal movements of the ITCZ were even more limited that no influence of either cold water masses during winter or warm subtropical water masses during summer are recognizable by a pronounced seasonal temperature difference in the Gulf of

Cadiz.

## 5 Conclusion

Two marine sediment cores from the Gulf of Cadiz and the Alboran Sea have been studied aiming at exploring the climatic variability with respect to precipitation and vegetation change in southern Iberia as well as resolving seasonal hydrological conditions during the mid- to late- Holocene. The following conclusions can be drawn from this study:

- Based on terrestrial n-alkane concentrations we found five major dry climate episodes in southern Iberia at 5.4 ± 0.3, from 5.1 to 4.9 ± 0.1, from 4.8 to 4.7 ± 0.1, from 4.4 to 4.3 ± 0.1 and, at 3.7 ± 0.1 cal. ka BP. These dry phases also impacted the vegetation, which suffered from environmental stress due to reduced water availability.

- The manifestation of the 4.2 ka event in southern Iberia appears to be more complex compared to other regions contrasting the idea of a prolonged dry episode at that time. Instead, our data suggests a shorter dry phase from 4.4 to 4.3 ± 0.1 cal. ka BP followed by a rapid shift towards wetter conditions at about 4.2 cal. ka BP, which lasted until approximately 3.8 cal. ka BP.

- SST reconstructions suggest fairly stable annual mean temperature conditions in the Alboran Sea and the Gulf of Cadiz with estimates values varying within the range of uncertainty of the calibration. Based on summer and winter SST estimates the new records imply different mechanisms driving the seasonal variability in these two oceanic basins. While in the Gulf of Cadiz opposite seasonal SST events (i.e. summer warming and winter cooling) of up to 2 °C amplitude enhanced the seasonal SST difference, only slight seasonal SST variations (i.e. summer cooling and winter warming) within the methodological uncertainty have been found in the Alboran Sea at different times.

- The new records further suggest that variability in precipitation and vegetation in southern Iberia and probably the hydrological variability in the Alboran Sea can be well explained by a NAO-like atmospheric circulation. Dominant positive NAO-like conditions appear contemporaneous with the observed dry phases over southern Iberia associated with slightly warmer annual mean SSTs and decreased MPP in the Alboran Sea.

- The variability of the seasonal SST contrasts in the Gulf of Cadiz seems to be closely related to North Atlantic Bond Events. We observe increasing seasonality with summer warming and winter cooling during Bond Events 3 and 4. We propose that during winter the southward transport of cold water masses from the North Atlantic affected the Gulf of Cadiz, while during summer warm water masses originating from the STG reached our study site. Bond Event 2 is not reflected by a larger difference in our seasonal SST reconstructions. This is likely due to decreasing summer insolation that also influences the seasonal movements of the subtropical Azores Front, which in the Gulf of Cadiz can also limit either the inflow of cold or warm water masses during winter and summer, respectively.

**Data availability.** The data reported in this paper are archived in Pangaea (www.pangaea.de).

**Supplement.** The supplement related to this article is available online at:

**Author contribution.** JS, the main author of this study (with contributions from all co-authors), performed the geochemical analyses for reconstruction of precipitation, vegetation and, alkenones-based temperature. JS also established the new age models for the two sediment cores. Planktic foraminiferal analyses for seasonal SST estimates were provided by MW. ES

compiled the calibration data sets for the SIMMAX MAT analyses and computed SIMMAX SST. TB was responsible for alkenone analysis and quality control.

**Competing interests.** The authors declare that they have no conflict of interest.

**Acknowledgments.** This research was performed in the framework of the CRC 1266 "Scales of transformation" (project number: 2901391021)  funded by the DFG (German Research Foundation). E. Salgueiro was funded by Fundação para a Ciência e Tecnologia (fellowship: SFRH/BPD/26525/2006 & SFRH/BPD/111433/2015). Sample material has been provided by the GeoB Core Repository and the IODP Core Repository at the MARUM – Center for Marine Environmental Sciences, University of Bremen, Germany. In this respect, kind support is acknowledged provided by J. Pätzold, V. B. Bender and, A. Wülbers. The authors acknowledge F. Burjachs for his kind data supply. We are also very grateful for enormous support from S. Koch with the geochemical analyses. Additionally, we thank I. Feeser for assistance with the Bayesian age modelling and W. Hamer for helping with the statistical analyses. We also thank the four anonymous referees as well as A. Garcia-Alix for their helpful comments on improving the manuscript.

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

**Figures and tables**

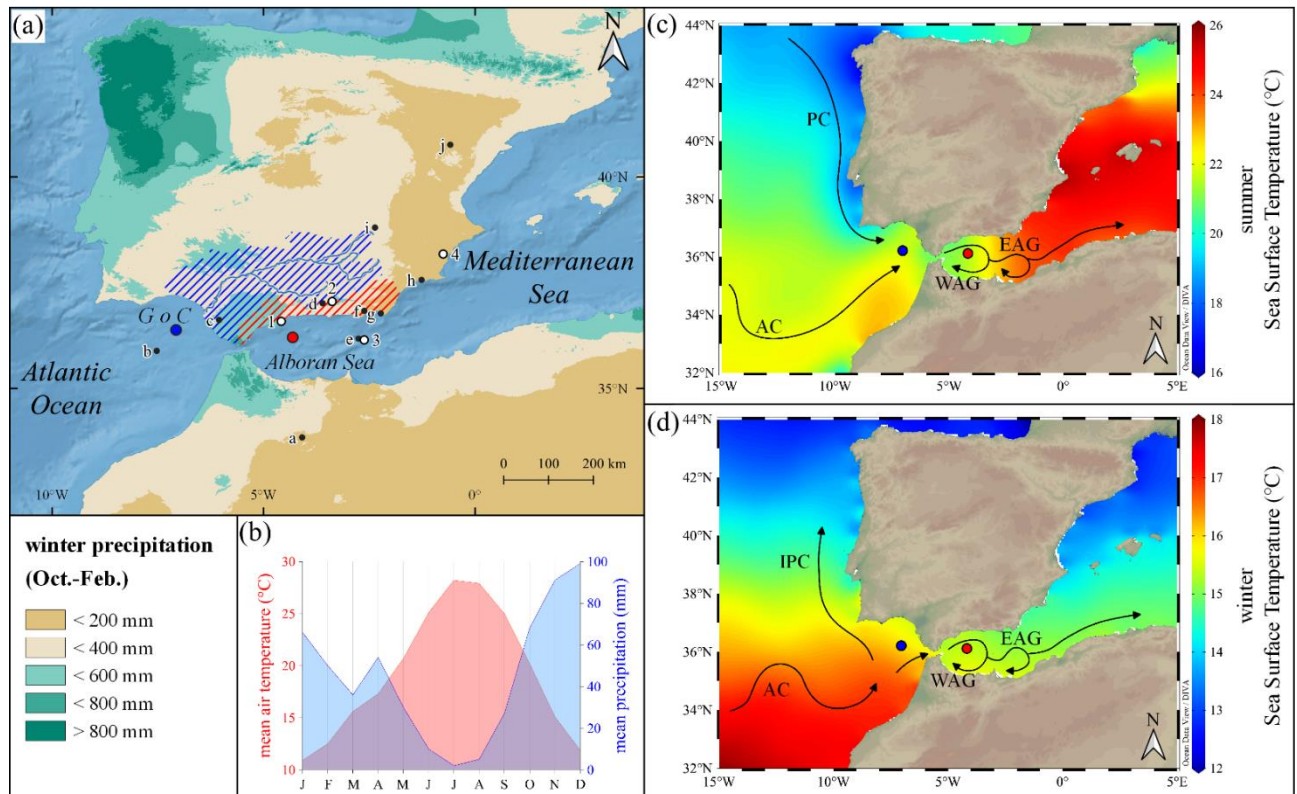

Figure 1: Overview maps. (a) overview with discussed (black dots) and shown (white dots) references at the Iberian Peninsula and studied sediment cores. Cores of this study: GeoB5901-2 (blue dot) and ODP-161-976A (red dot). Shown references: 1: El Refugio Cave (Walczak et al., 2015), 2: Borreguil de la Virgen and Borreguil de la Caldera (García-Alix et al., 2018), 3: MD95-2043 (Cacho et al., 2001; Fletcher and Sánchez Goñi, 2008; Martrat et al., 2014; Pérez-Folgado et al., 2003), 4: Elx (Burjachs and Expósito, 2015). Discussed references: a: Grotte de Piste (Wassenburg et al., 2016), b: MD99-2339 (Salgueiro et al., 2014), c: Lake Medina (Schröder et al., 2018), d: Padul (Ramos-Román et al., 2018a,b), e: TTR-293G (Rodrigo-Gámiz et al., 2014), f: San Rafael (Pantaléon-Cano et al., 2003), g: Cabo de Gata (Burjachs and Expósito, 2015), h: Mazarrón (Navarro-Hervás et al., 2014), i: Villaverde (Carrión et al., 2016) and, j: Ejulve cave (Moreno et al., 2017). The main river system and associated catchment of the Guadalquivir (hatched blue area) is shown as well as the river catchments of various small-scale rivers draining the southern Sierra Nevada area (hatched red area). The river catchments have been downloaded from the website of the European Environmental Agency. The colour shading indicates the mean winter precipitation (October to February) from 1970 to 2000 in the area. The data was provided by WorldClim V2 (Fick and Hijmans, 2017). (b) Annual average precipitation (blue) and air temperature (red) at Sevilla airport (1981-2010) show the high seasonality with rainy and cold winters and hot and dry summers. Data was provided by the Spanish State Meteorological Agency (AEMET). Mean summer (c) and winter (d) SSTs for the period 1955 – 2012 are shown. The data was provided by the World Ocean Atlas 2013 (Locarnini et al., 2013) and processed with Ocean Data View 5.1.2 (Schlitzer, 2018). Black arrows indicate the surface currents in the area including the gyres in the Alboran Sea. Blue and red dot mark locations of sediment cores GeoB5901-2 and ODP-161-976A, respectively. AC – Azores Current, IPC – Iberian Poleward Current, PC – Portugal Current, EAG – East Alboran Gyre, WAG – West Alboran Gyre.

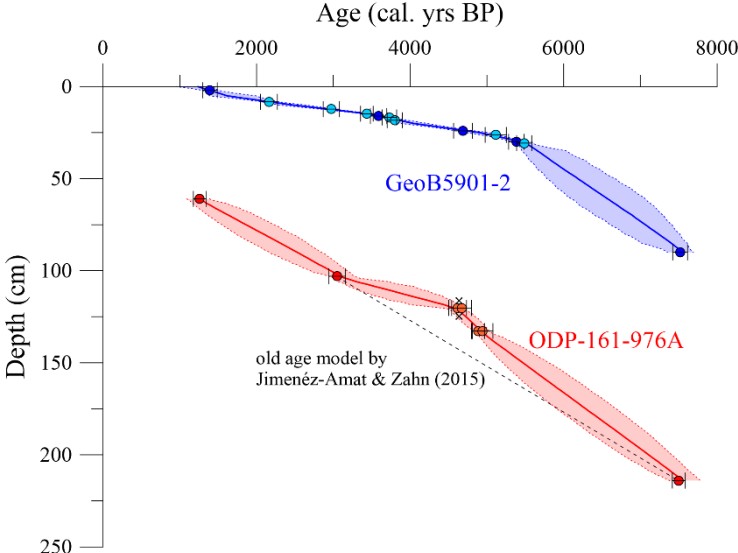

Figure 2: Age model. The age model from ODP-161-976A (red) is based on AMS[14]C dates (red dots) from Combourieu Nebout et al. (2002) and new AMS[14]C dates from this study (orange dots). Black crosses mark AMS[14]C dates considered as outliers. Black dotted line indicates the shift of the previous age model from Jiménez-Amat and Zahn (2015). The Age model of GeoB5901-2 (blue) is based on AMS[14]C datings done by Kim et al. (2004) (dark blue dots) and new AMS[14]C dates accomplished during this study (light blue dots). The red and blue shaded area indicates the 95% probability of the age model for ODP-161-976A and GeoB5901-2, respectively. All AMS[14]C dates are listed in Table 1.

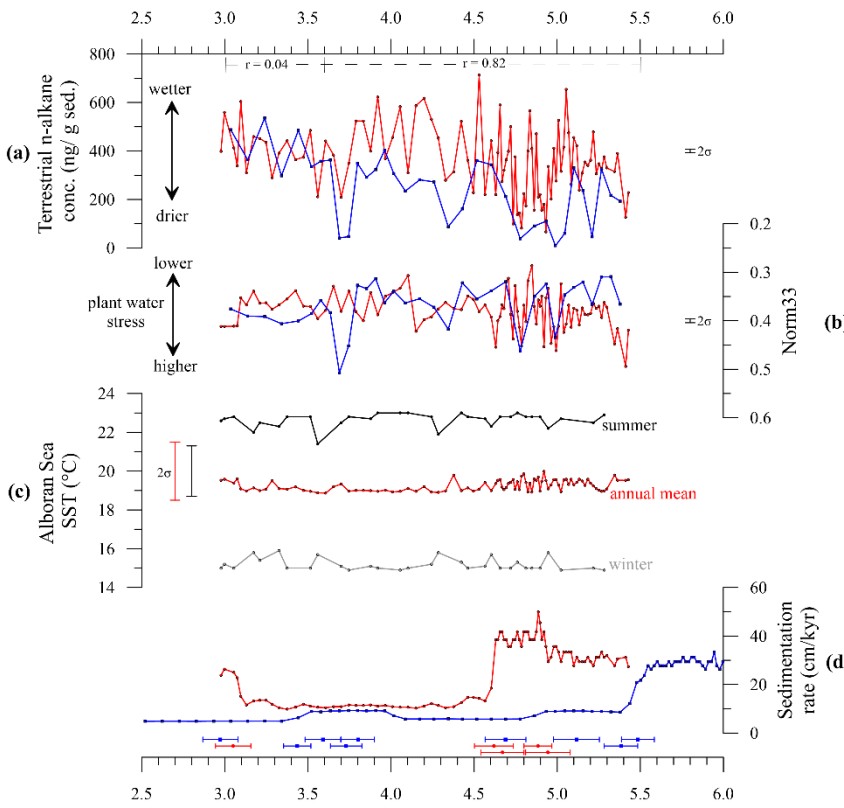

Figure 3: Data of marine sediment cores ODP-161-976A (red) and GeoB5901-2 (blue). (a) Terrestrial n-alkane concentration ($\Sigma C_{27-33}$) indicative of precipitation change. Pearson's correlation coefficient has been calculated based on 100-year time slices for two periods (see supplement for further information). (b) Norm33 n-alkane ratio showing plant water stress. (c) Alkenone and planktic foraminifera based Sea Surface Temperature (SST) reconstruction from ODP-161-976A (black – summer; grey – winter; red – annual mean). Error bars indicate the uncertainty of the calibration for the annual mean (red) and seasonal (black) reconstructions. (d) Sedimentation rate of both cores are shown for comparison. Coloured dots at the bottom show AMS[14]C age control points and associated 2σ errors.

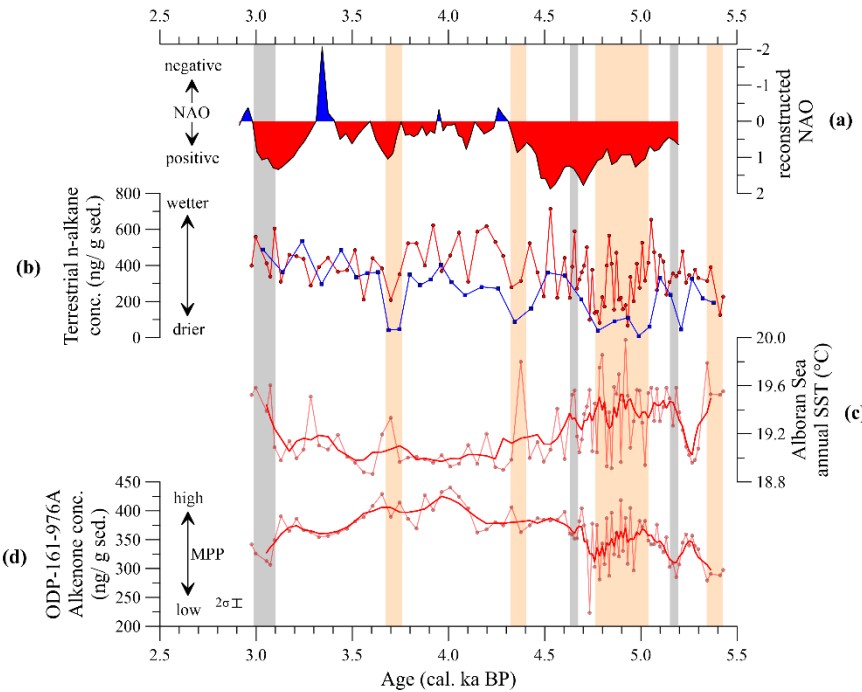

**Figure 4: Potential driver of atmospheric and Alboran Sea climate variability. (a) NAO reconstruction (Olsen et al., 2012). (b) Terrestrial n-alkane concentrations from ODP-161-976A (red) and GeoB5901-2 (blue) (this study). (c) Annual mean SST from ODP-161-976A (this study). (d) Alkenone concentration from ODP-161-976A indicative of MPP variability (this study). Thick red lines show the 5-point running means. Orange bars indicate dry phases observed in this study, while grey bars indicate periods of annual mean SST maxima and MPP minima at the core location of ODP-161-976A.**

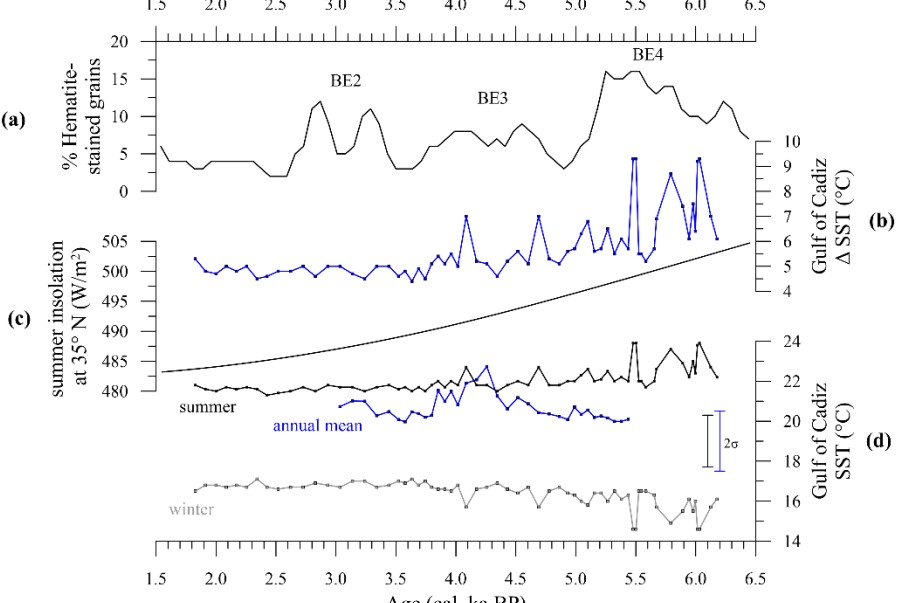

**Figure 5: Oceanic variability in the Gulf of Cadiz. (a) Stacked hematite-stained grain percentages from marine sediment cores in the North Atlantic showing Bond Events (BEs) 2 to 4 (Bond et al., 2001). (b) Seasonal SST difference (winter – summer) deduced from the GeoB5901-2 foraminiferal data. (c) Summer insolation at 35° N (Laskar et al., 2004). (d) Alkenone and planktic foraminifera based SST reconstruction from GeoB5901-2 (black – summer; grey – winter; blue – annual mean). Error bars indicate the uncertainty of the calibration for the annual mean (blue) and seasonal (black) reconstructions.**

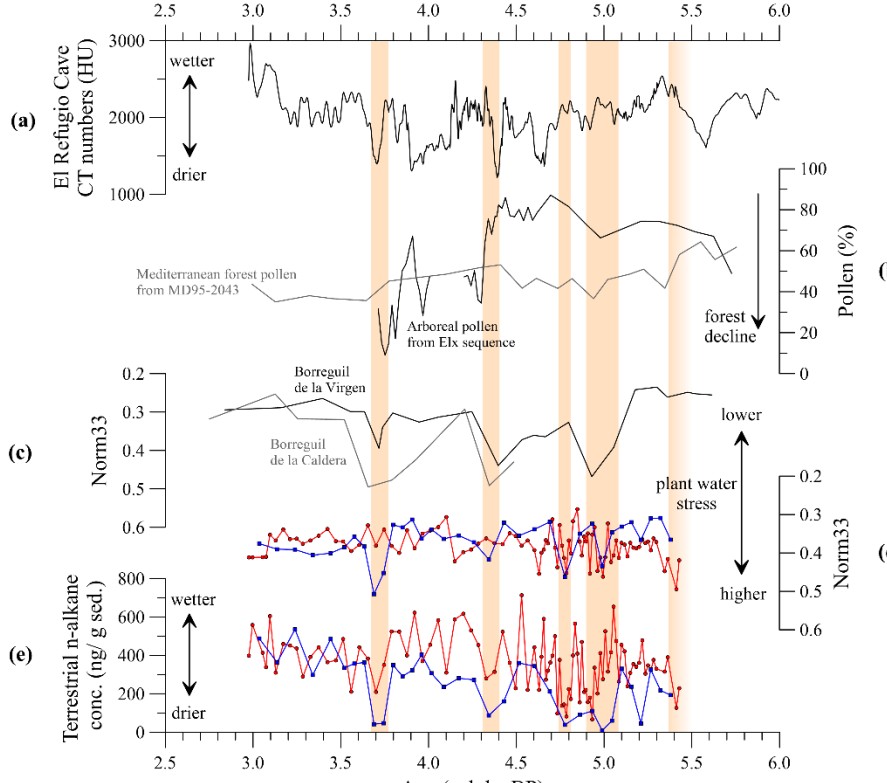

**Figure 6: Proxy data from the Iberian Peninsula. (a) Speleothem density data from El Refugio Cave (Walczak et al., 2015). (b) Pollen data from Elx sequence (Burjachs and Expósito, 2015; Burjachs et al., 1997) and marine sediment core MD95-2043 (Fletcher and Sánchez Goñi, 2008). (c) Norm33 n-alkane ratios from Borreguil de la Caldera and Borreguil de la Virgen. These data have been calculated on the basis of n-alkane raw data from García-Alix et al. (2018). (d) Norm33 n-alkane ratios from ODP-161-976A (red) and GeoB5901-2 (blue) (this study). (e) Terrestrial n-alkane concentrations from ODP-161-976A (red) and GeoB5901-2 (blue) (this study). The orange bars indicate dry phases observed in this study. The locations of all the references are shown in Figure 1.**


**Table 1: Age model of sediment cores ODP-161-976 and GeoB5901-2. All dates from previous studies have been re-calibrated. Not considered dates are shown in grey.**

| Sediment core | Lab. No. | Depth (cm) | AMS$^{14}$C age (yr BP) ±σ | Cal. age (yr BP) ±2σ | Dated material | Reference |
|---|---|---|---|---|---|---|
| GeoB5901-2 | KIA-14522 | 2.00 | 1840 ± 35 | 1392 ± 96 | planktic mix | Kim et al. (2004) |
| GeoB5901-2 | KIA-53006 | 8.25 | 2485 ± 25 | 2162 ± 110 | *G. ruber* w+p | this study |
| GeoB5901-2 | KIA-53005 | 12.25 | 3185 ± 27 | 2973 ± 106 | *G. ruber* w+p | this study |
| GeoB5901-2 | KIA-53002 | 14.75 | 3545 ± 26 | 3436 ± 82 | *G. ruber* w+p | this study |
| GeoB5901-2 | KIA-14521 | 16.00 | 3685 ± 35 | 3590 ± 108 | planktic mix | Kim et al. (2004) |
| GeoB5901-2 | KIA-53003 | 16.75 | 3789 ± 27 | 3730 ± 95 | *G. ruber* w+p | this study |
| GeoB5901-2 | KIA-53004 | 18.25 | 3852 ± 27 | 3801 ± 100 | *G. ruber* w+p | this study |
| GeoB5901-2 | KIA-14520 | 24.00 | 4500 ± 40 | 4689 ± 123 | planktic mix | Kim et al. (2004) |
| GeoB5901-2 | KIA-52665 | 26.25 | 4820 ± 35 | 5116 ± 138 | *G. ruber* w+p | this study |
| GeoB5901-2 | KIA-14518 | 30.00 | 5035 ± 40 | 5384 ± 101 | planktic mix | Kim et al. (2004) |
| GeoB5901-2 | KIA-52666 | 30.75 | 5130 ± 40 | 5486 ± 99 | *G. ruber* w+p | this study |
| GeoB5901-2 | KIA-14516 | 90.00 | 7035 ± 55 | 7518 ± 97 | planktic mix | Kim et al. (2004) |
| GeoB5901-2 | KIA-13704 | 120.00 | 7495 ± 50 | 7955 ± 121 | planktic mix | Kim et al. (2004) |
| ODP-161-976C | KIA-6435 | 61.00 | 1710 ± 40 | 1259 ± 82 | *G. bulloides* | Combourieu Nebout et al. (2002) |
| ODP-161-976C | KIA-6436 | 103.00 | 3235 ± 30 | 3050 ± 106 | *G. bulloides* | Combourieu Nebout et al. (2002) |
| ODP-161-976A | KIA-53326 | 116.25 | 4435 ± 35 | 4636 ± 138 | *G. bulloides* | this study |
| ODP-161-976A | KIA-53235 | 120.25 | 4435 ± 30 | 4619 ± 117 | *G. ruber* w+p | this study |
| ODP-161-976A | KIA-53327 | 120.25 | 4480 ± 40 | 4671 ± 130 | *G. bulloides* | this study |
| ODP-161-976A | KIA-53236 | 124.75 | 4435 ± 35 | 4636 ± 138 | *G. ruber* w+p | this study |
| ODP-161-976A | KIA-53234 | 132.75 | 4650 ± 35 | 4885 ± 84 | *G. ruber* w+p | this study |
| ODP-161-976A | KIA-53325 | 132.75 | 4700 ± 50 | 4945 ± 134 | *G. bulloides* | this study |
| ODP-161-976C | KIA-6437 | 214.00 | 7010 ± 50 | 7500 ± 84 | *G. bulloides* | Combourieu Nebout et al. (2002) |