# Peer review of "Multi-decadal atmospheric and marine climate variability in southern Iberia during the mid- to late- Holocene"

_Climate of the Past, 2018_

## Short Comment (SC1) · 22 Nov 2018

We have read this interesting paper about the mid-late Holocene climatic evolution in the western Mediterranean. However, we have observed that some previously published data have been taken/interpreted in a wrong way, more specifically the geochemical data from southern Spain. Authors took these data from a data repository / data descriptor paper (García-Alix et al., 2018; Pangaea), but they did not take into account the source paper where the specific explanation/interpretation of the data for this local area was provided (Garcia-Alix et al., 2017).

In the manuscript authors explained that (Page 6: lines 20-23):"The Norm33 ratio or the proportion of the C33 n-alkane homologue, respectively, is a function of a change

in C4 plant distribution according to air temperature and/ or precipitation change (Bush and McInerney, 2013; Herrmann et al., 2016; Leider et al., 2013; Rommerskirchen et al., 2006; Vogts et al., 2009; Vogts et al., 2012).". Afterwards they stated that (Page 9: lines 15-18): "Notably, in our records all observed drought episodes are paralleled by Norm33 maxima (Figure 4). These maxima indicate a shift towards a higher abundance of C4 plants, which are much more adapted to drier (and warmer) conditions (e.g. Bush and McInerney, 2013). Moreover, our data gains support from Sierra Nevada bog sediments Borreguil de la Virgen and Borreguil de la Caldera (Figure 4; García-Alix et al., 2018), which also indicate an increase in C4 plant abundance during the droughts".

This interpretation of the relationship between C4 and the Norm33 index (C33 vs C29 alkanes) is an overgeneralization, and more information about the local environments is needed to understand the meaning of n-alkanes in the different regions/areas, in particular in the Sierra Nevada (S Iberia). This information is available in a recent plant survey conducted in Sierra Nevada alpine wetlands (including Borreguil de la Virgen and Borrequil de la Caldera sites that authors mentioned). This survey was performed to understand the n-alkane (Garcia-Alix et al., 2017) and the organic  $\delta$ 13C meaning in local paleorecords (Jiménez-Moreno et al., 2013). Firstly, no native C4 plants are found in that area: "Vegetation of the catchment basin primarily consists of members of the Poaceae and Cyperaceae. The mean measured  $\delta$ 13C value of Poaceae and Cyperaceae is  $-29.3\pm1.4\%$  and  $-26.7\pm0.7\%$  respectively. By comparison, the carbon isotopic composition of the 16 plant samples (including Poaceae, Cyperaceae, Artemisia, Asteraceae, Fabaceae, Lamiaceae, Plantago, Ranunculaceae and an unidentified bryophyte) taken from near the lake ranges from -30.3% to -25.8% with a mean value of -27.0±1.4‰" (Jiménez-Moreno et al., 2013). The n-alkane signature of recent plants (C3 plants) is the same as the one of the paleorecords in these alpine areas: "Our modern plant and soil survey in the extreme Sierra Nevada environments (Fig. 1c) shows that the distance plants occur from a water source, such as wetlands, controls the length of n-alkanes carbon chains. In particular, plants that are in
or near the water pools show a stronger predominance of the shorter carbon chains." (Garcia-Alix et al., 2017). There is further information in the supplementary material of that paper. Therefore, the relationship between C33 and C29 alkanes, the Norm33 in Schirrmacher's draft, cannot be used as a C4 plant proxy in these sites (Borreguil de la Virgen and Borreguil de la Caldera): 1) because C4 plants do not occur in the area, and 2) because the alkane chain length is related to the plant water availability in these specific sites.

Anyway, the reason why these data seem to agree with authors' C4 reconstruction is because the Norm33 index in the Sierra Nevada alpine sites (Borreguil de la Virgen and Borreguil de la Caldera) is higher in dry periods, but not related to a higher C4 input. Therefore, Sierra Nevada data make sense for this paper, but their meaning should be related to the alternation of dry-wet periods, instead of C3-C4 plant fluctuations in these alpine sites.

Although we found evidence of north-Africa aeolian inputs in these alpine sites from southern Iberia based on inorganic geochemistry i.e: Zr/Al Zr/Th or Ca/Al proxies (Garcia-Alix et al., 2017; Jiménez-Espejo et al., 2014; Mesa-Fernández et al., 2018), we have not been able to identify a true evidence of C4 organic matter input in these alpine sites so far. The best way to identify C3-C4 plants is by using carbon isotopes, and as it was mentioned before, carbon isotopes in Holocene bulk sediments from these alpine sites show no clear signature of C4 plants (Garcia-Alix et al., 2017; García-Alix, Jiménez-Moreno, Anderson, Jiménez Espejo, & Delgado Huertas, 2012; Jiménez-Espejo et al., 2014; Jiménez-Moreno et al., 2013). The only potential C4 input in Sierra Nevada alpine wetlands may be at the YD-Holocene boundary (in Laguna de Río Seco), when the  $\delta$ 13C values of the organic matter from bulk sediments reached -18‰.This might be related to the release of the sediment stored in the YD glaciers during their melting (Jiménez-Espejo et al., 2014). Carbon isotopes in the n-alkanes from these sites are needed to get an accurate answer to this question.

A similar comment can be applied to (Page 9: lines 18-19): "In addition, pollen data
from Elx and Padul record an increase in C4 grasses (Poaceae) at ca. 5.4, 5.0, 4.8, 4.4 and, 3.7 ka BP (Burjachs pers. comm., 2018; Ramos-Román et al., 2018b". Ramos Roman's paper did not mention that Poaceae species from Padul were C4-Poaceae: actually C4-Poaceae have not been identified in this record so far.

In summary, n-alkanes from Borreguil de la Virgen, and Borreguil de la Caldera, along with Poaceae from Padul might be used for comparison as humidity/drought proxies, but not as C4-plant proxies in the mentioned sites from southern Spain.

New literature cited:

Garcia-Alix, A., Jimenez Espejo, F. J., Toney, J. L., Jiménez-Moreno, G., Ramos-Román, M. J., Anderson, R. S., . . . Kuroda, J. (2017). Alpine bogs of southern Spain show human-induced environmental change superimposed on long-term natural variations. Scientific Reports, 7, 7439 doi:https://doi.org/10.1038/s41598-017-07854-w García-Alix, A., Jiménez-Espejo, F. J., Jiménez-Moreno, G., Toney, J. L., Ramos-Román, M. J., Camuera, J., . . . Queralt, I. (2018). Holocene geochemical footprint from Semi-arid alpine wetlands in southern Spain. Scientific Data, 5, 180024. doi:10.1038/sdata.2018.24 García-Alix, A., Jiménez-Moreno, G., Anderson, R. S., Jiménez Espejo, F. J., & Delgado Huertas, A. (2012). Holocene environmental change in southern Spain deduced from the isotopic record of a highelevation wetland in Sierra Nevada. Journal of Paleolimnology, 48(3), 471-484. doi:https://doi.org/10.1007/s10933-012-9625-2 Jiménez-Espejo, F. J., García-Alix, A., Jiménez-Moreno, G., Rodrigo-Gámiz, M., Anderson, R. S., Rodríguez-Tovar, F. J., . Pardo-Igúzquiza, E. (2014). Saharan aeolian input and effective humidity variations over western Europe during the Holocene from a high altitude record. Chemical Geology, 374-375, 1-12. doi:https://doi.org/10.1016/j.chemgeo.2014.03.001 Jiménez-Moreno, G., García-Alix, A., Hernández-Corbalán, M. D., Anderson, R. S., & Delgado-Huertas, A. (2013). Vegetation, fire, climate and human disturbance history in the southwestern Mediterranean area during the late Holocene. Quaternary Research, 79(2), 110-122. doi:https://doi.org/10.1016/j.ygres.2012.11.008 Mesa-Fernández, J.

CPD
M., Jiménez-Moreno, G., Rodrigo-Gámiz, M., García-Alix, A., Jiménez-Espejo, F. J., Martínez-Ruiz, F., . . . Ramos-Román, M. J. (2018). Vegetation and geochemical responses to Holocene rapid climate change in the Sierra Nevada (southeastern Iberia): the Laguna Hondera record. Clim. Past, 14(11), 1687-1706. doi:10.5194/cp-14-1687-2018

---

## Referee Comment (RC1) · Anonymous Referee #1 · 29 Nov 2018

I find the manuscript from Schirrmacher of particular interest, especially for its combination of terrestrial and marine proxies in two close regions, which at the end show significant differences in the marine response to climatic changes. Of particular relevance to re-dating two records and giving a new age models, which produce high-resolution data. I have two general observations and several along the text. General observations 1) I think for a special issue dedicated to 4.2 ka event, it would be useful to expand the discussion on this point. The chronology suggested for this event is not really coincident with the interval where it is often found (see for instance for the Mediterranean Bini et al. this issue Climate of the past discussion). The range of ages found for this interval overlaps two dry event you find (i.e. 4.4 and 3.8 ka cal BP), but separated by an interval of wetter conditions. So the impression that this interval is much shorter

[Figure]
* * *
and/or older than in other records (but see also Isola et al. this issue or Kaniesky et al. this issue). For such reason your record is of particular interest because, it seems well dated and can give new perspectives on the topic. I think adding some sentences on these points would be very useful. 2) In many points the authors state that there is a good agreement between different proxies and different archives, when it is not always the case In some instances the authors recognize these discrepancies but in other not. I suggest some changes along the text. 3) Probably it would be useful to plot in figure 2 also the older ages model. In a way that older records will not be anymore selected if not recalculated on the new age models. Specif points

Pag. 2 line 4, just to note that the Neoglacial onset in the Apennine (central Italya) started at 4.2 too (Zanchetta et al., 2012 QR) Pag. 2 line 6: mean primary productivity (MPP), it is the first time quoted in the text. Pag. 3 line 2, delete regime after Mediterranean Pag. 3 line 7 mm instead of ml Pag. 3 line 14 delete during winter (written twice) Pag. 4 line 14 why you don't refer to Reimer et al. 2015? Pag. 4 lines 18-19 linearly interpolated. . ...be more precise which ages are interpolated. Pag. 6 line 2, 10 ccm? Do you mean 10 cm? or? Pag. 6 line 13 Proxy restrictions? What do yu mean precisely? Pag. 6 line 17 Jalali et al. 2016, 2017 Pag. 6 line 23 Vogts et al., 2009,2012 Pag. 7 line 18 how changes of 1 °C are significant considering the accuracy of the methods? Pag. 7 line 20 "Annual mean SST in GeoB5901-2 vary stable? Is very stable around ca. 20.0°C? It would be useful to give numbers as mean±sd. This can give also an idea of significant deviation from the mean. Pag. 8 line 11 . . ..well matches. . ..later you wrote it is not always the case. It is better to write some like "there is a general agreement. . .. Pag. 9 I think here to expand the discussion on 4.2 event is of particular interest. Pag. 9 line 15 "drought episodes are paralleled. . ." I think once again caution is necessary and description of mismatching is necessary. Pag. 9 lines 26-26. How is it possible that insolation decrease and SSt increase? Is there any wrong in this sentence? Or a further explanation is necessary? Which temperature are really recording your proxy? It probably needs some explanation. End pag. 9 beginning pag. 9. Please check carefully. It is little confusing and it is not always evident

to understand which SST you are referring to (mean, seasonal). Pag. 10 lines around 15, some further, more explicit comment on what temperature are measuring with your proxies is necessary. Pag. 10 lines 19-21. I don't think second decimal can be considered significant considering the age model. Pag. 10 lines 21-21. "These events, notably, differ from...." Surely this part needs to be expanded a little more..... Pag. 10 I have no particular problem abou the selection of Goslin et al. 2018 record, but there are also others. Is there any special reason? Is this record better dated? More robust? Pag. 10 line 26 "....compare well...." See previous observations.

Overall, the manuscript is well written and quite clear. Considering my general observations I think it needs moderate revision before acceptance. I recommend expanding the part related to 4.2 cal BP events.

---

## Referee Comment (RC2) · Anonymous Referee #2 · 5 Dec 2018

The overall quality of the paper is good; the paper addresses relevant scientific questions of the journal and presents new data. The text is mostly well written but lacks an into detail comparison to other records of the region as well as a detailed description of and introduction to the ocean currents around the Strait of Gibraltar and their evolution, which might be of great value related to the topic of the study. The section 4.2 sets it more or less in a regional context by the aid of regional to global phenomena, but is a bit exaggerated. E.g. I am not sure about the connection of Bond Events to the data of the study and about the substantial conclusions. The applied methods are clear, allow reproduction by fellow scientists and the authors give proper credit to related work and original data; assumptions are mostly valid as I see a difficulty in interpreting seasonal variability from the data, as to my opinion the sedimentation rate is too high as well

as the interval of sampling too big to write such a detailed interpretation). I am not convinced of the title, not about the "multi-decadal", nor about "southern Iberia". SST, maybe as well as Alboran sea and Gulf of Cadiz or oceanic variability should somehow be included in the title. The abstract is a bit excessive in its scientific statement, the overall presentation of the study is well structured and quite clear, however e.g. the beginning of the discussion is rather a results paragraph. Language is fluent and precise. Some figures should be clarified, references are ok in guantity and guality; naming of references within the text need to be checked for order. Section 1.1, line 5: mentioning of that figure is wrong, a precipitation curve would show the precipitation during winter e.g. Line 6: would be nice to see the Atlantic regime within the figure. Btw, you use ml in the text and mm as unit for precipitation in the figure, can you adjust that? Line 14: you mention again figure one, to my opinion in the wrong sentence. Concerning figure 1: (a) the figure shows too much of the Iberian Peninsula, you can easily reduce the area you show and exclude the Ejulve cave. A north arrow or coordinates are missing as well as a scale. The river beds could be shown more clearly. And I would not call the red shaded area the Alboran sea catchment, as a catchment should be related to the input area rather than the endmember of the area affected by the rivers (e.g. you call the other catchment Guadalquivir catchment, not Gulf of Cadiz catchment). I would also not use alphabetic letters for the discussed references, as it is difficult to read and find them within the caption, if you use a, b, c already for the subdivision of figure 1. January should not be written with capital letter in the caption. (b) and (c) could also be completed by coordinates or a north arrow and a scale. What are the white spots within (b) and (c)? Section 2, line 25: resampled on 0,5 cm is not wright, as you mention every second centimetre in line one of page 4. Section 2.2, Age model: line 19, 20: can you interpret the sedimentation rates by the use of other studies? Line 15: what is the reason for that massive shift? Can you explain that? Line 16, 17: the exclusions of the ages that you have is not really explained and the reason of lowest analytical error is not enough. Can you explain the "errors" in greater detail, where they might come from etc? Figure 2: why the abrupt steps of the sedimentation rate of ODP and smooth
increases and decreases of GEOB? Figure caption is very long, could you include the naming of the record within the figure next to the line? Section3, results: you do not include cal after the naming of an age, this is not consistent with the legends oif the axes of the figures. Section 2.3: why abbreviation of methanol MeOH? Looks like a molecular formula, which would be CH4O.. Page 7, Line 2: mentioning of figure 1 is not necessary. Page 8: line 7 and 8 is too my opinion exaggerated. Page 8, section 4, line 21+22: references are missing and included with more detailed information in line 1,2 and 3 on page 9, which could be included in page 8, line 21. Line 7 on page 8, rephrase "moreover, a forest..." as it is unclear. Line 15, drought episodes parallel to Norm 33... I don't think so! Line 23, where can I see that in figure 3? Page 11, line 15, why is there no explanation why bond 2 is not visible? Figure 6: not really discussed within the text. Section 5: the conclusion should be rephrased and maybe restructured too, some bullet points of your study, what is the most important interpretation etc.

---

## Referee Comment (RC3) · Anonymous Referee #3 · 10 Dec 2018

This paper presents two re-analysed cores from off the south coast of Iberia, one from the Gulf of Cadiz and the other from the Alboran Sea covering the mid-late Holocene. The authors suggest that, whilst close within a spatial context these two marine environments are influenced by quite different climate forcers. The paper is well laid out and nicely written, with detailed presentation of the identified wet-dry phases in the mid to late Holocene. Before recommending publication however, some significant areas of clarification are still required. My major concerns / points for consideration are listed first, followed by more minor typo style points and suggested any small changes to the Figures. I hope these comments prove useful to the authors. 1) The introduction seems to focus mainly on the 4.2ka event where the data set and the remainder of the paper is much broader than that one event. I would like to see an extension of

the introduction to cover more of the mid-late Holocene "events" in detail. The introduction could also include more background on the major forcing mechanisms in play here and discussed later in the paper. The NAO bit is in the study area section, but there is no info on the Bond events, and how these may influence ocean circulation and SST's in this region. 2) Figure 1 needs improvement. A scale, north arrow and labelling of the different oceans, countries etc is needed for a non local expert reader. 3) It would be nice to see the sedimentation rates through time plotted in your figures (Figure 3 for example) this would help get a feeling of how the different cores were deposited over time and will help inform the reader with regards to sample density vs time and therefore resolution of the data set. This is critical when interpreting changes at the multi-decadal level. 4) You state that two dates were removed (page 4 line 17) as they gave the same value as another date at 120cm. Can you explain why they were removed? Does this not just suggest a rapid accumulation rate over this period of the core and that all dates are valid? 5) I have some concerns about the use of the n-alkane data as the primary proxy for wetter or dryer conditions; I fear this proxy has been over extended in the interpretation. The areas I would like clarification are: a) it would be nice to see a couple of example chromatograms from the n-alkane work, especially during the extreme wet and dry periods (supplementary info is appropriate). This would help clarify if the material is originating from the same/similar source locations throughout the record. My concern is that over such a large catchment and long time period, changes in rainfall may be geographically heterogeneous, leading to the removal of organic matter from different parts of the catchment at different rates over time. b) Linked to this, more explanation of the physical mechanism of n-alkane removal by runoff is required (at least in your response to these comments). I'm concerned that under dry conditions, C3 dominated environments (forests for example) are less susceptible to water and sediment loss than C4 dominated environments, due to the physical make-up and bonding of their soils by root systems. If this is the case then the co-variation between n-alkane concentration reduction and "C4 proxy increase" is actually not showing an increase in C4 vegetation abundance within the catchment, but

a change in the relative loss of n-alkanes from each environment within the catchment. c) I would also direct the authors to the following paper, which suggests that identifying between C3 and C4 vegetation using n-alkanes is not straightforward, this needs consideration and clarification. Bush and McInerney (2013) Leaf wax n-alkane distributions in and across modern plants: Implications for paleoecology and chemotaxonomy. Geochimica et Cosmochimica Acta 117 (2013) 161–179. d) How can you be sure that reductions in the n-alkaline concentration in the core are not just a dilution effect from marine sediment deposition? Showing sediment accumulation rate on the same graph would clarify this (see comment above). 6) There are a few places in the text where you suggest this are "well correlated" or that events are well replicated in both cores (page 7 line11, page 8 line 11, page 10 line 7). I think the paper would benefit greatly from some stats to back up these statements, which are currently based on a visual assessment of the data. Being able to demonstrate a relationship between the cores will greatly enhance the robustness of the conclusions drawn. 7) I would also suggest that you investigate the periodicity of the wet-dry events shown in the record. NAO and bond events have well documented "frequencies", if you can demonstrate that these events in your record have a similar frequency this would again add weight to the argument that these major climate modes maybe the dominant mechanism controlling changes seen in your record. 8) If you are confident that the Norm33 does represent changes in C3-C4 vegetation distribution within the catchment (see comments above), I direct you to page 9 where you suggest that changes in vegetation community composition may change on very rapid time scales and that this can be accurately recorded within the ocean records. Under modern conditions, is there evidence of changes between C3 and C4 vegetation makeup over the time scales you see in the sediments? And, could you use these proxies to quantify the extent of vegetation change that would have been seen to get such a change in Norm33. What I'm asking I guess is, does the extent of Norm33 change seen in the record make sense, in terms of both rate of change from C3-C4 and the extent (%) of vegetation cover that would have to have changed, if contextualised by our understanding of the catchment under modern conditions? 9) The

SST data resolved from alkenone data is interesting but it must be clear within the text (when interpreting) and within the figures when this data falls within the 1°C error that you state in the methods (I suggest adding error bands in the figures). For example, in Figure 3 most of the peaks and troughs in your seasonal SST data are within error. This data is best interpreted in terms of differences between season temperatures (which you do well). Don't over interpret unless the max/min temps fall outside your (+/-?) 1°C error. 10) I think that more detail on the significance of not seeing Bond event 2 should be added. You suggest this is the first time higher seasonality in SST in relation to mid Holocene Bond events is described this far south. More mechanistic detail of how this N Atlantic process effects the Gulf of Cadiz would be helpful in understanding why there are difference between the mid and late Holocene Bond events and how it shows up at your sites. More minor comments: a) Page 3 line 14, "during winter" repeated. b) Page 6 line 1, "cm" not "ccm" c) Page 7 lines 1 and 2, use "in high resolution" not "on high resolution". d) Page 10 line 8, Figure 6 is introduced into the text before Figure 5, re-order figures. e) Figure 1 needs changes listed above

---

## Referee Comment (RC4) · Anonymous Referee #4 · 11 Jan 2019

Schirrmacher et al. present the results of a reanalysis of two marine cores from southern margin of Iberia spanning the late Holocene and discuss the possible factors and mechanism behind the reconstructed climate variability, with a focus on the 4.2 ka BP event. While not new - provided that the chronological uncertainties are better constrained - the results of the study could improve our understanding of the spatial characterization of the climatic conditions during the 4.2 ka BP event, especially as both summer and winter temperatures are reconstructed. However, I find the discussion of the data rather superficial and the discussion of the mechanisms minimal. While I support the publication of the paper, several issues, detailed below, needs to be addressed. 1) Chronology and resolution. While it is claimed that the records have high-resolution, this is not evident from the data. It is not clear how many samples have been analyzed,

especially for core 976A and what was the resolution: 0.5, 2 cm? Further, the choice for excluding several of the data points from the final age-depth model seem to be arbitrary – the exclusion of the ages with lower precision lead to further exclusions. How would the age-depth model have been if the samples with the lower precision were kept, instead (±10 years at 4000 cal BP does not make a big difference). Further, the choice of linear interpolation has been shown to give less reliable ages (Blaauw et al., 2018). Why not using Bayesian modeling? 2) The "results" and "discussions" chapters should be better separated, some of the text under the later would better fit under the former. 3) Their seem to be multiple issues with the "alignment" of the proxies, possibly resulting form the less precise (see above) chronology. Which of the several periods is identified precisely with the 4. 2ka event? Further, given that both summer and winter temperatures are reconstructed, the discussion should be separated for the two seasons. Next, rather than assuming that the 4.2 ka event was dry in the region and try to support this by choosing one or other of the "peaks" in the data I suggest starting with multiple hypothesis and discuss them in light of your data. Several studies in the wider study region have shown that the 4.2 ka BP event could have ben wet (e.g., Zielhofer et al., 2018) during winter. 4) The mechanisms described in chapter 4.2 ("Possible drivers. . .) rely more on Ausin et al. (2015) than on the data from the power. See also the comment above and the detailed comments below and try to improve the interpretation by providing a mechanistic evidence for the described processes.

Minor comments P1, L23: Dansgaard et al (1993) is outdated, perhaps some newer and better references would be better P2, L2: numerous other events are not resolved in NGRIP. . . P2, L12-13. I am not an archaeologist/historian, but perhaps "turnover" is not the best word to be used in this context P2, L20. Please detail the contrast P3. The word "relatively' is overused in the chapter 1.1. While Iberia is relatively cool (L2) compared to N Africa, is relatively hot, compared to N Canada. Please give the values for the temperature, it would allow readers to better understand the present-day climatic conditions. P3, L7 you mean mm instead of ml P3, L15: please detail the circulation, separately for the season, it is not clear from the text (e.g., you discuss low SST in

the Atlantic margin and than jump to warm inflow to the Alborean Sea...) Materials and methods: please improve the description of the sampling strategy, it is not clear what resolution you achieved in the end. Age model: see the comments above, the choices need to be better explained. A critical discussion on how a different choice of exclusions would have affected the results would be welcomed. P6, l1: ccm is cm3? P6, proxy reconstructions. Please give values for the Q for both rivers, as well as for the seasonal discharges to better understand the seasonality of alkanes in the cores P7, results: please ad "cal" after ka (e.g., 4.3 ka cal BP) P7, L11: the contemporaneity should be discussed in the light of chronological issues P7 and 8, results: the entire chapter is somewhat confusing, please try to simplify it. Also, it is not clear how the various dry/cold/warm periods have been found to be contemporaneous. P8, l19: was it dry in winter or summer? See the detailed comment above P9, L11: 20 years..what is the age error here? P9, L15: winter or summer, again? Generally (I repeat myself) the discussion should be clearly separated for summer and winter P9, L25-26: not clear, the cooling trend would result in colder, not warmer SSTs P9, L25: "at that time" What time? P9, L29 and next lines on P10: for which period are these temperatures given? P10, L9: hm, the resolution problem. Was it high or low? My quick calculations show that the resolution is closer to 100 years at the time of interest.... P10, l15-19: for which period does this paragraph refer to? Generally, chapter 4.1 is a mix of results and discussion, most of it should go under "results" P10, chapter 42.. This is the "meat" of the paper, but the discussion is quite weak. I also think that "NAO-like variability" is quite over abused. Further, if the ANO is to be used, perhaps it would be more useful to use a NAO reconstruction, rather than a storminess one, which could result from other factors than NAO (e.g., Olsen et al., 2012) P11, L15: the comparison with the IRD record is useful as long as the mechanisms linking the two are better described. Else, correlation and causality are different. Please improve the discussion by including mechanistic explanation that could result in the variability described here.

References Blaauw, M., Christen, J. A., Bennett, K. D., and Reimer, P. J.: Double the dates and go for Bayes — Impacts of model choice, dating density and quality on chronologies, Quaternary Science Reviews, 188, 58-66, https://doi.org/10.1016/j.quascirev.2018.03.032, 2018.

Olsen, J., Anderson, N. J., and Knudsen, M. F.: Variability of the North Atlantic Oscillation over the past 5,200 years, Nature Geosci, 5, 808-812, http://www.nature.com/ngeo/journal/v5/n11/abs/ngeo1589.html#supplementary-information, 2012.

Zielhofer, C., Köhler, A., Mischke, S., Benkaddour, A., Mikdad, A., and Fletcher, W. J.: Western Mediterranean hydro-climatic consequences of Holocene iceberg advances (Bond events), Clim. Past Discuss., 2018, 1-20, 10.5194/cp-2018-97, 2018.

---

## Author Comment (AC1) · 12 Feb 2019

Dear Antonio García-Alix,

Thank you very much for your important comment! Also we want to apologize that we obviously did not read your 2017-paper carefully and interpreted your data in the wrong way!
Concerning the interpretation within this paper we will follow your suggestions and interpret the Norm33 fluctuations as consequence of plant water stress rather than C4-C3 variability.

Kind regards

---

## Author Comment (AC2) · 12 Feb 2019

Dear Referee #1,

Thank you very much for your comments on our manuscript. Please find below some replies to your comments.

**Comment:** I think for a special issue dedicated to 4.2 ka event, it would be useful to expand the discussion on this point.
**Reply:** We agree that the discussion on the actual 4.2 ka event is short. We did not want to put the discussion of the 4.2 ka event in the centre of the manuscript since the events observed at ca. 5.0 ka and 3.8 ka seem to be much more pronounced in our records. Nonetheless, a more detailed discussion on the timing of the 4.2 ka event in our records has been added.

**Comment:** Sometimes the text contains statements about good agreement between different proxies and archives while in some instances differences are recognized.
**Reply:** We computed correlation analysis (Pearson) with our records underscoring the good agreement in general. Moreover, the differences, which occur, are more critically discussed.

**Comment:** Probably it would be useful to plot in figure 2 also the older ages model. In a way that older records will not be any more selected if not recalculated on the new age models.
**Reply:** We agree. The previous age model of Jiménez-Amat and Zahn (2015) has been indicated with a dotted line in Figure 2.

**Comment:** Pag. 2 line 6: mean primary productivity (MPP), it is the first time quoted in the text.
**Reply:** You probably refer to line 26. Please note that MPP is already introduced in line 24.

**Comment:** Pag. 3 line 2, delete regime after Mediterranean
**Reply:** 'regime' has been deleted.

**Comment:** Pag. 3 line 7 mm instead of ml
**Reply:** 'mm' has been written instead of 'ml'

**Comment:** Pag. 3 line 14 delete during winter (written twice)
**Reply:** the second 'winter' has been deleted

**Comment:** Pag. 4 line 14 why you don't refer to Reimer et al. 2015? Pag. 4 lines 18-19 linearly interpolated.....be more precise which ages are interpolated.
**Reply:** Also following Referee #4 comments we used Bayesian Modelling to construct our age model. Accordingly, the whole chapter has been re-written, but we now referred to Reimer et al. (2013) mentioning the marine13 calibration curve and the 400 yr reservoir correction. We hope that Referee #1 intentionally meant this paper, because we are not aware of any paper from Reimer et al. in 2015.

**Comment:** Pag. 6 line 2, 10 ccm? Do you mean 10 cm? or?
**Reply:** We meant 'cubic centimetres' and adjusted it towards 'cm$^3$'.

**Comment:** Pag. 6 line 13 Proxy restrictions? What do you mean precisely?
**Reply:** With 'restrictions' we mean the (spatial, temporal, etc.) limits of the used proxies. But we re-structured also this chapter and included it into the discussion.

**Comment:** Pag. 6 line 17 Jalali et al. 2016, 2017 Pag. 6 line 23 Vogts et al.,2009,2012
**Reply:** Thanks, we adjusted this. Unfortunately, this was a formatting problem with the citation software.

**Comment:** Pag. 7 line 18 how changes of 1∘C are significant considering the accuracy of the methods?
**Reply:** Of course, these changes are not significant because they are within the methodological error. To emphasize this, we inserted a statement that these SST changes need to be considered with caution. Additionally, we have added an error bar indicating the methodological error within Figure 3.

**Comment:** Pag. 7 line 20 "Annual mean SST in GeoB5901-2 vary stable? Is very stable around ca. 20.0∘C? It would be useful to give numbers as mean±sd. This can give also an idea of significant deviation from the mean.
**Reply:** We followed this suggestion.

**Comment:** Pag. 8 line 11 ....well matches....later you wrote it is not always the case. It is better to write some like "there is a general agreement. and Pag. 9 line 15 "drought episodes are paralleled" I think once again caution is necessary and description of mismatching is necessary.
**Reply:** You are right. We computed correlation factors (Pearson) also referring to a comment from Referee #3 in order to statistically support our statements.

**Comment:** Pag. 9 lines 26-26. How is it possible that insolation decrease and SSt increase? Is there any wrong in this sentence? Or a further explanation is necessary? Which temperature are really recording your proxy? It probably needs some explanation.
**Reply:** Sorry this was a mistake. We meant that over the Holocene SSTs are generally cooling (due to decreasing insolation) with lowest SSTs (in the mean) during the present. Consequently, it is reasonable that the SST in the Mid- Holocene was warmer compared to modern temperatures. We corrected this mistake.

**Comment:** End pag. 9 beginning pag. 9. Please check carefully. It is little confusing and it is not always evident to understand which SST you are referring to (mean, seasonal).
**Reply:** We agree that this is a little confusing and difficult to read. We re-structured the discussion chapter into terrestrial and marine sub-chapters and discuss the annual, winter and, summer SSTs more separately within the marine sub-chapter.

**Comment:** Pag. 10 lines around 15, some further, more explicit comment on what temperature are measuring with your proxies is necessary.
**Reply:** Comparing the Gulf of Cadiz alkenone (annual mean) SST with other data, we wrote: "*The here reconstructed annual mean SSTs appear actually more close to summer conditions*". To be more specific a more detailed discussion on this issue has been added.

**Comment:** Pag. 10 lines 19-21. I don't think second decimal can be considered significant considering the age model.
**Reply:** We aimed to delete every second decimal before submission, but obviously we must have missed those numbers. This has been changed.

**Comment:** Pag. 10 lines 21-21. "These events, notably, differ from...." Surely this part needs to be expanded a little more.....
**Reply:** We agree. Some additional sentences for explanation and discussion on this point were added.

**Comment:** Pag. 10 I have no particular problem about the selection of Goslin et al. 2018 record, but there are also others. Is there any special reason? Is this record better dated? More robust?

**Reply:** We followed the comment of Referee #4 and compared our data to the NAO reconstruction of Olsen et al. (2012). Initially, we chose the Goslin et al. (2018) record because it covers the whole time interval (until 5.5 ka BP), while the NAO-reconstruction from Olsen et al. (2012) ends at 5.2 ka BP.

---

## Author Comment (AC3) · 12 Feb 2019

Dear Referee #2,

Thank you very much for your comments! Please find some replies below.

**Comment:** The text is mostly well written but lacks an into detail comparison to other records of the region as well as a detailed description of and introduction to the ocean currents around the Strait of Gibraltar and their evolution, which might be of great value related to the topic of the study.
**Reply:** We agree. A more detailed comparison to key references such as El Refugio Cave as well as an introduction and discussion on the ocean currents has been added.

**Comment:** I am not convinced of the title, not about the "multi-decadal", nor about "southern Iberia". SST, maybe as well as Alboran sea and Gulf of Cadiz or oceanic variability should somehow be included in the title.
**Reply:** We think that the temporal resolution of our records is allowing multi-decadal resolution and is one of the key features of this study. Thus, we want to highlight this also in the title. However, Referee #2 is right with his comment that we do not resolve seasonal variations on multi-decadal resolution. "Southern Iberia" is the region the terrestrial plant proxies stem from and as our study is very local also with respect to the oceanic proxies we should keep this phrase. Nonetheless, we agree that the title so far excludes the oceanic regions, so we will include a "atmospheric and oceanic climate variability" in the title.

**Comment:** Section 1.1, line 5: mentioning of that figure is wrong, a precipitation curve would show the precipitation during winter
**Reply:** Correct! We have included a precipitation and atmospheric temperature curve in Figure 1.

**Comment:** e.g. Line 6: would be nice to see the Atlantic regime within the figure. Btw, you use ml in the text and mm as unit for precipitation in the figure, can you adjust that?
**Reply:** The regimes can be defined by the precipitation amount, therefore in a way it is already shown. Drawing an additional line etc. to our opinion would result in a too busy figure. The units have been adjusted.

**Comment:** Line 14: you mention again figure one, to my opinion in the wrong sentence.
**Reply:** Figure 1 has been adjusted so that winter precipitation (October to February) is shown. We think referring to Figure 1 is correct in this sentence.

**Comment:** Concerning figure 1: (a) the figure shows too much of the Iberian Peninsula, you can easily reduce the area you show and exclude the Ejulve cave. A north arrow or coordinates are missing as well as a scale. The river beds could be shown more clearly. And I would not call the red shaded area the Alboran sea catchment, as a catchment should be related to the input area rather than the endmember of the area affected by the rivers (e.g. you call the other catchment Guadalquivir catchment, not Gulf of Cadiz catchment). I would also not use alphabetic letters for the discussed references, as it is difficult to read and find them within the caption, if you use a, b, c already for the subdivision of figure 1. January should not be written with capital letter in the caption. (b) and (c) could also be completed by coordinates or a north arrow and a scale. What are the white spots within (b) and (c)?
**Reply:** Thanks a lot to Referee #2 for these helpful comments on Figure 1. Much of these have been adjusted! We think we should not reduce the size of the map by cutting off the northern part, because we want to highlight that the modern true moist Atlantic regime is far to the north while the area under consideration is a much dryer system. North arrow, scale and, coordinates have been added. We also tried to make the river beds more visual by adding a shading to the river bed.

"Alboran Sea catchment" has been renamed to "catchments from various small-scale rivers draining the southern Sierra Nevada". The naming of the catchments was also moved from the Figure into the figure caption. We agreed to differentiate between "shown" and "mentioned" sites because otherwise, we might imply wrong expectations to the reader. For that purpose, we used numbers and small letters for separation. We will keep this because capital letters, for example, would highlight these references too much. Furthermore, the use of small letters for figure subdivision is wanted by the journal. But, we think that in the final manuscript the small letters used for subdivision of the Figure 1 will be shown in bold, so that they should be easier to distinguish. As already said "January" has been replaced by "winter" anyway. The respective months are mentioned in addition. The white spots in subfigures (b) and (c) -now (c) and (d)- are mapping gaps. These are mainly occurring in coastal areas as well as in case of lakes. In order to remove some of the white spots we now show the elevation on land instead of continent in grey.

**Comment:** Section 2, line 25: resampled on 0,5 cm is not wright, as you mention every second centimetre in line one of page 4. Section 2.2,
**Reply:** We agreed that the description is difficult to read. We adjusted that in the following way: "*Sediment core ODP-161-976A (36°12.32' N; 4°18.76' W; 1108 m water depth) was retrieved in the Alboran Sea during JOIDES Resolution cruise in 1995 (Comas et al., 1996). To achieve multi-decadal resolution, the section from 100.0 cm to 149.0 cm was continuously sampled at 0.5 cm distances in the IODP Core Repository at MARUM in Bremen (Germany).*" For simplification we do not mention the two different sampling steps, which are the reason for the different temporal resolution of the geochemical and foraminiferal data.

**Comment:** Age model: line 19, 20: can you interpret the sedimentation rates by the use of other studies? Figure 2: why the abrupt steps of the sedimentation rate of ODP and smooth increases and decreases of GEOB? Figure caption is very long, could you include the naming of the record within the figure next to the line?
**Reply:** Unfortunately, comparable cores (timing, sampling resolution, location) are very scarce (see discussion of the oceanic variability) not allowing the comparison of sedimentation rates. Moreover, the interpretation of sedimentation rates –especially without any reference data- is difficult and beyond the scope of this study. Following a comment from Referee #4, we used Bayesian modelling to create the age models, which also resulted in more smoothed sedimentation rates except for one abrupt change in each sediment core. We re-wrote the figure caption and included the names of the sediment cores into the Figure.

**Comment:** Line 15: what is the reason for that massive shift? Can you explain that?
**Reply:** So far, we cannot decide whether the previous dates are "wrong" or "right". A measurement at the same sampling depths would be needed to do so, but this is impossible since these samples do not exist anymore. We assume that the shift is probably a consequence of a sampling of a much larger depth increment for dating by the previous studies.

**Comment:** Line 16, 17: the exclusions of the ages that you have is not really explained and the reason of lowest analytical error is not enough. Can you explain the "errors" in greater detail, where they might come from etc?
**Reply:** The errors are the methodological error from the dating itself and the calibration curve of course. Following Referee #4 we modelled the age model using a Bayesian approach (see comment above). Doing so, we kept all double-dated samples. For the new age model, we just excluded two AMS dates at 116.25 and 124.75 cm, because we assume a rather smooth and constant sedimentation rate. This is because there are no evidences for bioturbation etc. resulting in a 10 cm

thick section of similar age. Since the sample at 120.25 cm is also double-dated we just kept this date. This explanation has been added in the revised version.

**Comment:** Section3, results: you do not include cal after the naming of an age, this is not consistent with the legends of the axes of the figures.
**Reply:** This is true! We have adjusted that.

**Comment:** Section 2.3: why abbreviation of methanol MeOH? Looks like a molecular formula, which would be CH4O..
**Reply:** Correct! We deleted this lab-internal abbreviation since it is not used somewhere else in the text.

**Comment:** Page 7, Line 2: mentioning of figure 1 is not necessary.
**Reply:** We deleted the hint to Figure 1 here.

**Comment:** Page 8: line 7 and 8 is too my opinion exaggerated.
**Reply:** We have deleted this sentence.

**Comment:** Page 8, section 4,line 21+22: references are missing and included with more detailed information in line 1,2 and 3 on page 9, which could be included in page 8, line 21.
**Reply:** This was meant to be an introductory sentence for the following discussion, but we deleted it.

**Comment:** Line 7 on page 8, rephrase "moreover, a forest ..." as it is unclear.
**Reply:** We rephrased this sentence to *"Furthermore, a drastic forest opening in SE Iberia is indicated from the Elx pollen sequence at ca. 4.3 cal. ka BP and at Cabo de Gata around 4.4 cal. ka BP (**Fehler! Verweisquelle konnte nicht gefunden werden.**; Burjachs and Expósito, 2015)"*.

**Comment:** Line 15, drought episodes parallel to Norm 33... I don't think so!
**Reply:** The term "all" is replaced by "most" since we agree that in ODP-161-976A the last two periods of drought at ca. 3.8 and 4.3 ka BP are not accompanied by clear Norm33 peaks.

**Comment:** Line 23, where can I see that in figure 3?
**Reply:** We deleted the reference to Figure 3 here.

**Comment:** Page 11, line 15, why is there no explanation why bond 2 is not visible?
**Reply:** We agree. A critical and detailed discussion on Bond Event 2 has been added.

**Comment:** Figure 6: not really discussed within the text.
**Reply:** We have added a more detailed discussion on the mechanism proposed by Figure 6 and, more general, thriving the oceanic variability in relation to the Bond Events.

**Comment:** Section 5: the conclusion should be rephrased and maybe restructured too, some bullet points of your study, what is the most important interpretation etc.
**Reply:** We rephrased the "conlusion" trying to focus more on the most important outcomes of our study.

---

## Author Comment (AC4) · 12 Feb 2019

Dear Referee #3,

Thank you very much for your inspiring and helpful comments. Please find some replies below.

**Comment:** The introduction seems to focus mainly on the 4.2ka event where the data set and the remainder of the paper is much broader than that one event. I would like to see an extension of the introduction to cover more of the mid-late Holocene "events" in detail. The introduction could also include more background on the major forcing mechanisms in play here and discussed later in the paper. The NAO bit is in the study area section, but there is no info on the Bond events, and how these may influence ocean circulation and SST's in this region.
**Reply:** A true point. We restructured the introduction also explaining the NAO and the Bond Events in general. Nonetheless the focus on the 4.2 ka event as an example remained since we also followed the suggestion of Referee #1 and concentrate more on the 4.2 ka event in the discussion.

**Comment:** Figure 1 needs improvement. A scale, north arrow and labelling of the different oceans, countries etc is needed for a non local expert reader.
**Reply:** We have implemented a scale as well as a north arrow. Labelling of the Atlantic Ocean and Mediterranean Sea as well as the sub-basins such as the Gulf of Cadiz and the Alboran Sea is now included. We won't label countries for political reasons. Also the journal encourages us not to do so.

**Comment:** It would be nice to see the sedimentation rates through time plotted in your figures (Figure 3 for example) this would help get a feeling of how the different cores were deposited over time and will help inform the reader with regards to sample density vs time and therefore resolution of the data set. This is critical when interpreting changes at the multi-decadal level.
**Reply:** A very good point! We moved the sedimentation rates from Figure 2 to Figure 3.

**Comment:** You state that two dates were removed (page 4 line 17) as they gave the same value as another date at 120cm. Can you explain why they were removed? Does this not just suggest a rapid accumulation rate over this period of the core and that all dates are valid?
**Reply:** We extended the discussion on the age model and explain the removal of particular dates in more detail. Considering these two dates as evidence for an extraordinary high accumulation interval is ruled out, because in the sediment core itself there is no evidence by any lithological feature, grain-size or other properties for such anomaly. In the meantime, we also followed the suggestion of Referee #4 and modelled the age models using a Bayesian approach, that allows to better consider the reliability of these dates.

**Comment:** I have some concerns about the use of the n-alkane data as the primary proxy for wetter or dryer conditions; I fear this proxy has been over extended in the interpretation. The areas I would like clarification are: a) it would be nice to see a couple of example chromatograms from the n-alkane work, especially during the extreme wet and dry periods (supplementary info is appropriate). This would help clarify if the material is originating from the same/similar source locations throughout the record. My concern is that over such a large catchment and long time period, changes in rainfall may be geographically heterogeneous, leading to the removal of organic matter from different parts of the catchment at different rates over time.
**Reply:** We included example chromatograms in the supplement and additionally collected information on the vegetation of the catchment areas in order to assess this point.

**Comment:** b) Linked to this, more explanation of the physical mechanism of n-alkane removal by runoff is required (at least in your response to these comments). I'm concerned that under dry conditions, C3 dominated environments (forests for example) are less susceptible to water and sediment loss than C4 dominated environments, due to the physical make-up and bonding of their soils by root systems. If this is the case then the co-variation between n-alkane concentration

reduction and "C4 proxy increase" is actually not showing an increase in C4 vegetation abundance within the catchment, but a change in the relative loss of n-alkanes from each environment within the catchment. c) I would also direct the authors to the following paper, which suggests that identifying between C3 and C4 vegetation using n-alkanes is not straightforward, this needs consideration and clarification. Bush and McInerney (2013) Leaf wax n-alkane distributions in and across modern plants: Implications for paleoecology and chemotaxonomy. Geochimica et Cosmochimica Acta 117 (2013) 161–179.

**Reply:** We added a few sentences on the physical mechanisms in the discussion. Moreover, according to Short Report #1 the interpretation concerning the vegetation shifts will be less strong in the revised version. A large scale shift from C3 to C4 vegetation is not considered anymore, which would also reduce the concern expressed by Referee #3.

**Comment:** d) How can you be sure that reductions in the n-alkaline concentration in the core are not just a dilution effect from marine sediment deposition? Showing sediment accumulation rate on the same graph would clarify this (see comment above).

**Reply:** The sedimentation rate is plotted in Figure 3 to allow comparison (see reply above). Also we added a small discussion on whether the sedimentation rate affected the used proxies or not.

**Comment:** There are a few places in the text where you suggest this are "well correlated" or that events are well replicated in both cores (page 7 line11, page 8 line 11, page 10 line 7). I think the paper would benefit greatly from some stats to back up these statements, which are currently based on a visual assessment of the data. Being able to demonstrate a relationship between the cores will greatly enhance the robustness of the conclusions drawn.

**Reply:** We fully agree. We now calculated Pearson's correlation coefficients for compared data. A detailed description of how this has been done has been added to the supplement.

**Comment:** I would also suggest that you investigate the periodicity of the wet-dry events shown in the record. NAO and bond events have well documented "frequencies", if you can demonstrate that these events in your record have a similar frequency this would again add weight to the argument that these major climate modes maybe the dominant mechanism controlling changes seen in your record.

**Reply:** We tried to apply spectral analysis to the proxy records. Unfortunately, the analysed time period is too short for producing robust results, at least for the ca. 1500-year Bond cycle. Therefore, we refrain from applying this approach and have to limit the discussion on visual correlation.

**Comment:** If you are confident that the Norm33 does represent changes in C3-C4 vegetation distribution within the catchment (see comments above), I direct you to page 9 where you suggest that changes in vegetation community composition may change on very rapid time scales and that this can be accurately recorded within the ocean records. Under modern conditions, is there evidence of changes between C3 and C4 vegetation makeup over the time scales you see in the sediments? And, could you use these proxies to quantify the extent of vegetation change that would have been seen to get such a change in Norm33. What I'm asking I guess is, does the extent of Norm33 change seen in the record make sense, in terms of both rate of change from C3-C4 and the extent (%) of vegetation cover that would have to have changed, if contextualised by our understanding of the catchment under modern conditions?

**Reply:** As stated above the interpretation concerning C3-C4 vegetation change has been discarded.

**Comment:** The SST data resolved from alkenone data is interesting but it must be clear within the text (when interpreting) and within the figures when this data falls within the 1◦C error that you state in the methods (I suggest adding error bands in the figures). For example, in Figure 3 most of the peaks and troughs in your seasonal SST data are within error. This data is best interpreted in terms of differences between season temperatures (which you do well). Don't over interpret unless the max/min temps fall outside your (+/-?) 1◦C error.

**Reply:** We added a statement on the SST error in the discussion chapter and in order to visualize the error in the figures we plotted a single error bar displaying the mean analytical and/or methodological error. Unfortunately, the plotting of error bands, as suggested by Referee #3 would results in too busy figures, which would hamper the visual correlation.

**Comment:** I think that more detail on the significance of not seeing Bond event 2 should be added. You suggest this is the first time higher seasonality in SST in relation to mid Holocene Bond events is described this far south. More mechanistic detail of how this N Atlantic process effects the Gulf of Cadiz would be helpful in understanding why there are difference between the mid and late Holocene Bond events and how it shows up at your sites.

**Reply:** We agree. We extended the discussion on the Bond Events and associated mechanisms.

**Comment:** Page 3 line 14, "during winter" repeated.
**Reply:** This has been adjusted.

**Comment:** Page 6 line 1, "cm" not "ccm"
**Reply:** It has been changed to "$cm^3$" since we meant cubic centimetres.

**Comment:** Page 7 lines 1 and 2, use "in high resolution" not "on high resolution".
**Reply:** This has been changed.

**Comment:** Page 10 line 8, Figure 6 is introduced into the text before Figure 5, re-order figures.
**Reply:** Figures 5 and 6 have been completely re-structured.

---

## Author Comment (AC5) · 12 Feb 2019

Dear Referee #4,

Thank you very much for your helpful comments! Please find replies below.

**Comment:** 1) Chronology and resolution. While it is claimed that the records have high-resolution, this is not evident from the data. It is not clear how many samples have been analyzed, especially for core 976A and what was the resolution: 0.5, 2 cm? Further, the choice for excluding several of the data points from the final age-depth model seem to be arbitrary – the exclusion of the ages with lower precision lead to further exclusions. How would the age-depth model have been if the samples with the lower precision were kept, instead (±10 years at 4000 cal BP does not make a big difference). Further, the choice of linear interpolation has been shown to give less reliable ages (Blaauw et al., 2018). Why not using Bayesian modeling?

**Reply:** The comments on the chronology and resolution are greatly acknowledged! We have added a clear statement that the sampling resolution is 0.5 cm. Moreover, we followed the suggestion and used Bayesian modelling in order to construct our age model. Thereby, we kept the double dated samples (the ones formerly excluded due to lower precision) and just neglected two dates. The exclusion of these two dates is now explained on the basis of the lithology of the sediment core, which provides no evidence for an extraordinary high accumulation interval.

**Comment:** 2) The "results" and "discussions" chapters should be better separated, some of the text under the later would better fit under the former.

**Reply:** We have shifted the comparison of our records into the "results" chapter.

**Comment:** 3) Their seem to be multiple issues with the "alignment" of the proxies, possibly resulting form the less precise (see above) chronology. Which of the several periods is identified precisely with the 4.2ka event? Further, given that both summer and winter temperatures are reconstructed, the discussion should be separated for the two seasons. Next, rather than assuming that the 4.2 ka event was dry in the region and try to support this by choosing one or other of the "peaks" in the data I suggest starting with multiple hypothesis and discuss them in light of your data. Several studies in the wider study region have shown that the 4.2 ka BP event could have ben wet (e.g., Zielhofer et al., 2018) during winter.

**Reply:** For the "alignment" of the proxies we now used statistical methods (Pearson's correlation coefficient) to underline our interpretation. Also following Referee #1, we have emphasized the discussion on the 4.2 ka event. Thereby, we followed the suggestion of Referee #4 and started with multiple hypothesis. Furthermore, to improve the readability we divided the discussion on the SSTs for summer and winter season also following a comment from Referee #1.

**Comment:** 4) The mechanisms described in chapter 4.2 ("Possible drivers...) rely more on Ausin et al. (2015) than on the data from the power. See also the comment above and the detailed comments below and try to improve the interpretation by providing a mechanistic evidence for the described processes.

**Reply:** We increased the discussion on the driving mechanisms also providing more detailed ideas on how they work and can affect our data.

**Comment:** P1, L23: Dansgaard et al (1993) is outdated, perhaps some newer and better references would be better

**Reply:** We replaced Dansgaard et al. (1993) by Rasmussen et al. (2014).

**Comment:** P2, L2: numerous other events are not resolved in NGRIP...

**Reply:** We have deleted this reference.

**Comment:** P2, L12-13. I am not an archaeologist/historian, but perhaps "turnover" is not the best word to be used in this context
**Reply:** This part of the introduction has been deleted. Following Referee #3 we wanted to minimize the focus on the 4.2 ka event in the introduction. The introduction now focusses just on the climate and the mechanisms considered as possible driver.

**Comment:** P2, L20. Please detail the contrast
**Reply:** We added some examples for this contrast.

**Comment:** P3. The word "relatively' is overused in the chapter 1.1. While Iberia is relatively cool (L2) compared to N Africa, is relatively hot, compared to N Canada. Please give the values for the temperature, it would allow readers to better understand the present-day climatic conditions.
**Reply:** We added modern values for temperatures discussed in this chapter.

**Comment:** P3, L7 you mean mm instead of ml
**Reply:** We have corrected this.

**Comment:** P3, L15:  please detail the circulation, separately for the season, it is not clear from the text (e.g., you discuss low SST in the Atlantic margin and than jump to warm inflow to the Alborean Sea…)
**Reply:** We added a detailed discussion on the ocean circulation also separated for summer and winter season. The oceanic currents and circulation pathways have also been drawn into Figure 1.

**Comment:** Materials and methods:  please improve the description of the sampling strategy, it is not clear what resolution you achieved in the end. Age model:  see the comments above, the choices need to be better explained. A critical discussion on how a different choice of exclusions would have affected the results would be welcomed.
**Reply:** We re-wrote these chapters (see reply above).

**Comment:** P6, l1:  ccm is cm3?
**Reply:** Yes, it should be. We have adjusted it.

**Comment:** P6, proxy reconstructions. Please give values for the Q for both rivers, as well as for the seasonal discharges to better understand the seasonality of alkanes in the cores
**Reply:** We added seasonal discharge data for the Guadalquivir into the "study area" section. Unfortunately, we did not find such data for any rivers draining the southern Sierra Nevada. All we found were very recent data, which are affected by river dams and, thus, show a very different anthropogenic signal.

**Comment:** P7, results: please ad "cal" after ka (e.g., 4.3 ka cal BP)
**Reply:** We have adjusted this for the whole manuscript.

**Comment:** P7, L11: the contemporaneity should be discussed in the light of chronological issues; P7 and 8, results:  the entire chapter is somewhat confusing, please try to simplify it.  Also, it is not clear how the various dry/cold/warm periods have been found to be contemporaneous.
**Reply:** We re-structured the "results"-chapter and separated it for both sediment cores – each with a clear separation of terrestrial and oceanic proxy results. Afterwards, we added a sub-chapter

comparing the data of both sediment cores using also statistical approaches (see reply above) and also discussing chronological issues.

**Comment:** P8, l19: was it dry in winter or summer? See the detailed comment above
**Reply:** It was most likely dry in winter since our n-alkane proxy is probably biased towards the winter season. But, we have added this also in the discussion when discussing our n-alkane data.

**Comment:** P9, L11: 20 years..what is the age error here?
**Reply:** We deleted every second decimal within the ages. Also, according to the new age model this period is now longer. Furthermore, we gave the age uncertainty for every dry event observed in our study.

**Comment:** P9, L15: winter or summer, again? Generally (I repeat myself) the discussion should be clearly separated for summer and winter
**Reply:** We agree that this was confusing and difficult to read. We re-structured the discussion chapter into terrestrial and marine sub-chapters and, further, discussed the seasonal SST variations separately.

**Comment:** P9, L25-26:  not clear, the cooling trend would result in colder, not warmer SSTs
**Reply:** We meant that over the Holocene SSTs are generally cooling (due to decreasing insolation) with lowest mean SSTs during more recent times. Consequently, it is reasonable that the SST in the Mid- Holocene was warmer compared to today. We have restructured this sentence in order to make this clearer.

**Comment:** P9, L25: "at that time" What time?
**Reply:** We replaced "at that time" with "during the studied period".

**Comment:** P9, L29 and next lines on P10: for which period are these temperatures given?
**Reply:** The temperatures mentioned all focus on the studied period between 2.9 and 5.4 ka BP. We have added a clarification in this part.

**Comment:** P10, L9: hm, the resolution problem.  Was it high or low?  My quick calculations show that the resolution is closer to 100 years at the time of interest....
**Reply:** We included the resolution of the "low resolution" studies from the area for better comparison.

**Comment:** P10, l15-19:  for which period does this paragraph refer to?
**Reply:** For the whole analysed period. We added a statement to make this clear.

**Comment:** Generally, chapter 4.1 is a mix of results and discussion, most of it should go under "results"
**Reply:** We have adjusted this (see also reply above).

**Comment:** P10, chapter 42.. This is the "meat" of the paper, but the discussion is quite weak. I also think that "NAO-like variability" is quite over abused. Further, if the ANO is to be used, perhaps it would be more useful to use a NAO reconstruction, rather than a storminess one, which could result from other factors than NAO (e.g., Olsen et al., 2012)

**Reply:** We emphasized the discussion on the possible drivers. We also followed the suggestion by Referee #4 and now refer to the NAO reconstruction from Olsen et al. (2012). We, intentionally chose the Goslin et al. (2018) data because it covers the whole time period of our study.

**Comment:** P11, L15: the comparison with the IRD record is useful as long as the mechanisms linking the two are better described. Else, correlation and causality are different. Please improve the discussion by including mechanistic explanation that could result in the variability described here.
**Reply:** We introduced the Bond Events and associated changes broadly in the introduction and, further, improved the discussion on the mechanisms in the light of our data.